# Differential Privacy for Transformer Embeddings with Nonparametric Variational Information Bottleneck

## Abstract

We propose a privacy-preserving method for sharing text data by sharing noisy versions of their transformer embeddings. It has been shown that hidden representations learned by deep models can encode sensitive information from the input, making it possible for adversaries to recover the input data with considerable accuracy. This problem is exacerbated in transformer embeddings because they consist of multiple vectors, one per token. To mitigate this risk, we propose Nonparametric Variational Differential Privacy (NVDP), which ensures both useful data sharing and strong privacy protection. We take a differential privacy (DP) approach, integrating a nonparametric variational information bottleneck (NVIB) layer into the transformer architecture to inject noise into its multi-vector embeddings and thereby hide information, and measuring privacy protection with Rényi divergence (RD) and its corresponding Bayesian Differential Privacy (BDP) guarantee. Training the NVIB layer calibrates the noise level according to the utility of the downstream task. We test NVDP on the General Language Understanding Evaluation (GLUE) benchmark and show that varying the noise level gives us a useful trade-off between privacy and accuracy. With lower noise levels, our model maintains high accuracy while offering strong privacy guarantees, effectively balancing privacy and utility.

## 1 Introduction

Deep learning methods are highly dependent on the availability of data, but sharing data is often limited by privacy concerns. This is especially true for text data, where private attributes such as gender or age are mixed with useful information and can be detected by attackers (Li et al., 2018). Even when sharing the embeddings of text instead, these representations can still contain sensitive information, and an adversary could use techniques like a GAN (Generative Adversarial Network) attack (Hitaj et al., 2017) to reverse-engineer the original input, potentially reconstructing the text and exposing private information. This is especially true for state-of-the-art attention-based models like transformers, where a text embedding consists of many vectors, one per text token. We want to be able to share such transformer embeddings while still addressing privacy concerns.

Differential Privacy (DP) (Dwork, 2006) is widely recognized as the benchmark for rigorously quantifying and mitigating privacy risks associated with processing sensitive data. However, integrating DP into machine learning (Abadi et al., 2016; Bassily et al., 2014) remains a persistent challenge, primarily due to the substantial drop in model performance compared to non-private versions.

We take the approach of applying DP at the stage of sharing the data, before applying machine learning. This approach has the advantage that the shared data can be reused for multiple purposes and to train multiple models. However, DP relies on noise to remove information, and many types of data, in particular discrete data such as text, do not have a straightforward model of noise. If we first embed the data with a transformer encoder and then share noisy embeddings, then we can apply this privatization approach to any data which can be embedded with a transformer. The transformer architecture has gained prominence for its flexibility and effectiveness across diverse tasks, in particular text.

In this paper, we propose a method for adding noise to transformer embeddings which results in both retaining useful information and DP guarantees. The key to this method is using a nonparametric variational information bottleneck (NVIB) regularizer to train a noise model which is calibrated to the downstream task. Our proposed nonparametric variational differential privacy (NVDP) model first uses NVIB to train a distribution over transformer embeddings, and then samples from this distribution to get a noisy embedding which can be shared. Unlike prior approaches to local DP for sentence embeddings (Du et al., 2023; Meehan et al., 2022) that rely on task-agnostic noise mechanisms, our NVDP model's primary novelty lies in its task-aware calibration of randomness. In addition, NVIB calibrates the total randomness across the multiple vectors in a document's transformer embedding, rather than adding noise independently to individual sentence embeddings. Using Rényi Divergence (RD) as a measure of privacy (Geumlek et al., 2017), we show that NVDP provides an effective trade-off between level of privacy and usefulness in the downstream task.

This paper makes the following contributions:

- Propose a DP model (NVDP) which uses a variational information bottleneck (VIB) regularizer to balance DP against task-aware usefulness.
- Propose to use a nonparametric VIB regularizer (NVIB) to support sharing multi-vector attention-based representations with DP.
- Show empirically that NVDP provides a useful trade-off between privacy and utility.
- Show empirically that NVIB regularization is more effective than VIB regularization used in an analogous way.

## 2 Background

Our proposed method builds on previous work on Rényi differential privacy and NVIB.

### 2.1 Differential Privacy

DP has emerged as the leading standard for rigorously addressing privacy leakage during the processing of sensitive datasets (Dwork & Roth, 2014). It prevents attackers from deducing too much information about the input data solely from the algorithm's outputs.[1] There are two primary settings for DP: global differential privacy (GDP) and local differential privacy (LDP). LDP is designed for scenarios where data comes from end users who do not trust a central data collector or third parties with their raw data (Dwork & Roth, 2014). Under LDP, each user independently perturbs their data before sharing it, ensuring that sensitive information remains protected even if the data collector is compromised. By introducing noise or randomness at the user level, LDP enables the collection of aggregated insights from a dataset without revealing individual-level details, thus maintaining privacy while still allowing for statistical analysis (Dwork & Roth, 2014; Abadi et al., 2016). In our work, we aim to protect embeddings of text by sharing noisy embeddings, and thus we apply LDP. To formalize this, DP ensures that for every pair of adjacent inputs $\boldsymbol{x}$ and $\boldsymbol{x}'$ and a randomized algorithm $M$, the distribution of $M(\boldsymbol{x})$ and $M(\boldsymbol{x}')$ is close to each other. Originally, the distributions of $M(\boldsymbol{x})$ and $M(\boldsymbol{x}')$ are considered adjacent if they differ by all the attributes of a single record, but in practice the notion of adjacency can vary. For example, Sun et al. (2019) defines two sentences as adjacent if they differ by at most 5 consecutive words, where each sentence is split into segments of 5 words. In our framework, we operate under the standard LDP model, where the guarantee concerns the indistinguishability of the mechanism's output for any two distinct inputs $\boldsymbol{x}$ and $\boldsymbol{x}'$. Thus, we consider all inputs $\boldsymbol{x}$ to be neighbors of all other inputs $\boldsymbol{x}'$.

**Rényi Differential Privacy (RDP)** is a variation of standard $(\epsilon, \delta)$-DP that employs RD as a metric to measure the distance between the output distributions $Q$ of $M(\boldsymbol{x})$ and $Q'$ of $M(\boldsymbol{x}')$ (Geumlek et al., 2017). This approach is especially advantageous for training machine learning models that require DP.

---

[1]There is no distinction between sensitive information and non-sensitive information, since we can't tell with certainty what an attacker can deduce from the available information. Thus DP simply tries to reduce all information.

**Definition 2.1.** Given two probability distributions $\mathcal{Q}$ and $\mathcal{Q}'$ defined over domain $\mathcal{Z}$, the RD of order $\lambda > 1$ is:

$$D_\lambda(Q||Q') = \frac{1}{\lambda - 1} \log \left( \int_{\boldsymbol{z}} Q(\boldsymbol{z}) \left( \frac{Q(\boldsymbol{z})}{Q'(\boldsymbol{z})} \right)^{\lambda - 1} \right) \tag{1}$$

Using this divergence, we can formally define the privacy guarantee for a mechanism. In our work, we focus on the local privacy setting where the inputs are individual data points.

**Definition 2.2** $((\lambda, \epsilon)$-RDP). A randomized mechanism $M : \mathcal{X} \to \mathcal{Z}$ satisfies $(\lambda, \epsilon)$-Local RDP if for any pair of adjacent inputs $\boldsymbol{x}, \boldsymbol{x}' \in \mathcal{X}$, the following holds:

$$D_\lambda(M(\boldsymbol{x})||M(\boldsymbol{x}')) \leq \epsilon \tag{2}$$

The privacy parameter $\epsilon$, often called privacy budget, measures the privacy loss associated by the output of the algorithm: $\epsilon = 0$ provides perfect privacy, meaning the output is completely independent of the input. As $\epsilon \to \inf$, the privacy guarantees diminish, offering no protection for the input data. The other parameter $\lambda$ controls the importance of the worst-case privacy violations, reducing to Kullback–Leibler (KL) divergence at its lowest value and the maximum log-difference at its highest value.

**Bayesian Differential Privacy (BDP)** offers an alternative interpretation of privacy by focusing on the change in an adversary's belief. As introduced by Triastcyn & Faltings (2020), instead of comparing two adjacent inputs $\boldsymbol{x}$ and $\boldsymbol{x}'$, BDP analyzes how an output from a mechanism $M(\boldsymbol{x})$ changes an adversary's posterior belief about the input $\boldsymbol{x}$ compared to their prior belief. This prior is typically formed from knowledge of the underlying data distribution, which we can denote as $\mathcal{X}$. Formally, this is captured in the $(\epsilon_\mu, \delta_\mu)$-BDP definition:

**Definition 2.3.** Let $M : \mathcal{X} \to \mathcal{Z}$ be a randomized mechanism applied to individual data points. Then $M$ satisfies $(\epsilon_\mu, \delta_\mu)$-Local BDP if, for all $\boldsymbol{x}$, a data point $\boldsymbol{x}'$ drawn from the data distribution $\boldsymbol{x}' \sim \mathcal{X}$, and all measurable subsets $S \subseteq \mathcal{Z}$, the following holds:

$$\Pr[M(\boldsymbol{x}) \in S] \leq e^{\epsilon_\mu} \Pr[M(\boldsymbol{x}') \in S] + \delta_\mu \tag{3}$$

Here, $\Pr[M(\boldsymbol{x}') \in S]$ marginalizes over both the outputs in $S$ and the datapoints $\boldsymbol{x}' \sim \mathcal{X}$, and $\delta_\mu$ accounts not only for the probability of catastrophic privacy loss from the mechanism itself but also incorporates the uncertainty an analyst has about the true data distribution. This framework provides a practical way to reason about privacy for typical data points, which is highly relevant for machine learning models trained on specific data domains.

**Combining BDP and RDP** Triastcyn & Faltings (2020) also points out that RD can be used to define an upper bound on the worst-case privacy loss over sampled outputs. They use this fact to define a privacy accountant framework which allows multiple accesses to the data, but it can also be used to simply define a new privacy measure which is less sensitive to worst-case loss and closer to expected-case loss over outputs, depending on the setting of $\lambda$. This gives us a privacy measure which summarises the privacy risk over both the distribution of alternative examples $x'$ and the possible outputs. In our experiments, we leverage this connection by first calculating the RD and then converting it into an interpretable $(\epsilon_\mu, \delta_\mu)$-BDP guarantee. This conversion is applied to a single instance of the stochastic mechanism. We use the formulation detailed in Theorem 2 of Triastcyn & Faltings (2020). For the full derivation and implementation details of this measure, we refer the reader to the original work by Triastcyn & Faltings (2020).

### 2.2 Nonparametric Variational Information Bottleneck

NVIB (Henderson & Fehr, 2023) is a VIB regularizer for attention layers. It replaces the set of key vectors in a transformer's attention layer with a latent mixture of impulse distributions, specified as a set of vectors $\boldsymbol{Z}$ and their normalized weights $\boldsymbol{\pi}$. To access this mixture distribution, it generalizes the attention function to "denoising attention". It then uses Bayesian nonparametrics to define prior and posterior distributions over these mixture distributions, and a KL divergence with the prior to regularize the amount of information in its posterior.

**Prior** Since NVIB uses Bayesian inference to specify its posterior, it needs a prior over the latent space. Since the number of keys in an attention layer grows with the length of the input and the input can be arbitrarily large, this prior needs to specify a distribution over sets of weighted vectors ($\boldsymbol{\pi} \in \mathbb{R}^m, \boldsymbol{Z} \in \mathbb{R}^{m \times d}$) which have no finite limit to their size $m$. Nevertheless, we can still define probability distributions over this infinite set by applying Bayesian nonparametric methods. In particular, Henderson & Fehr (2023) use a Dirichlet Process to define the prior distribution $\text{DP}(G_0^p, \alpha_0^p)$ over unboundedly large mixtures ("p" designates the prior and "q" designates the posterior). The vectors $\boldsymbol{Z}_i$ are generated by the base distribution $G_0^p$, which is a Gaussian distribution with parameters ($\boldsymbol{\mu^p} = \boldsymbol{0}, (\boldsymbol{\sigma^p})^2 = \boldsymbol{1}$). The weights $\boldsymbol{\pi}$ are generated with a symmetric Dirichlet distribution parameterized by the total pseudo-count $\alpha_0^p = 1$, which tends to generate large weights on a few vectors and a long tail of exponentially smaller weights.

**Posterior** An NVIB layer uses Bayesian inference to parameterize its posterior distribution $\text{DP}(G_0^q, \alpha_0^q)$ in terms of a set of pseudo-observations computed from the set of vectors which the transformer inputs to the layer. These input vectors $\boldsymbol{x} \in \mathbb{R}^{n \times d}$, where $n$ is the number of vectors in the sequence and $d$ is the dimensionality of each vector, are each individually projected to a pseudo-count $\alpha_i^q \in \mathbb{R}$ and a pair of Gaussian parameters ($\boldsymbol{\mu_i^q} \in \mathbb{R}^d, (\boldsymbol{\sigma_i^q})^2 \in \mathbb{R}^d$). The base distribution $G_0^q$ is a mixture of Gaussians, where each pair of Gaussian parameters specifies a component of the mixture, and the pseudo-counts specify their weights. In addition, there is an $n+1^{\text{th}}$ component of the base distribution specified by the prior's parameters, so $(\alpha_{n+1}^q, \boldsymbol{\mu_{n+1}^q}, (\boldsymbol{\sigma_{n+1}^q})^2) = (\alpha_0^p, \boldsymbol{\mu^p}, (\boldsymbol{\sigma^p})^2)$.

$$F \sim \text{DP}\left(G_0^q, \ \alpha_0^q\right) \qquad \alpha_0^q = \sum_{i=1}^{n+1} \alpha_i^q \qquad G_0^q = \sum_{i=1}^{n+1} \frac{\alpha_i^q}{\alpha_0^q} G_i^q \qquad G_i^q = \mathcal{N}(\boldsymbol{\mu_i^q}, \boldsymbol{I}(\boldsymbol{\sigma_i^q})^2) \qquad (4)$$

As in previous work (Fehr & Henderson, 2024), during training our NVIB layer approximates samples of weighted vectors from this posterior as $\boldsymbol{\pi} \sim \text{Dir}(\boldsymbol{\alpha^q})$ and $\boldsymbol{Z_i} \sim \mathcal{N}(\boldsymbol{\mu_i^q}, (\boldsymbol{\sigma_i^q})^2)$, where $\text{Dir}(\cdot)$ is a Dirichlet distribution. An NVIB layer also has the capability to drop specific embeddings by setting their pseudo-counts to zero, effectively reducing the complexity of the representation.

**NVIB training objective** The NVIB loss regularizes the flow of information through the latent representation. The loss consists of three components: a task loss ($L_T$) and two KL divergence terms, $L_D$ and $L_G$. The $L_T$ term serves as a supervised learning objective, guiding the latent representation to retain enough information to perform the task. The $L_G$ term encourages noise in the Gaussian components, making them each less informative. The $L_D$ term both encourages noise in the weights and encourages some of the Dirichlet distribution's pseudo-counts to reach zero, effectively eliminating some vectors and reducing the overall information capacity of the latent representation. Two hyperparameters, $\lambda_D$ and $\lambda_G$[2], are added to adjust the impact of both the $L_D$ and $L_G$ regularization terms:

$$L = L_T + \lambda_D L_D + \lambda_G L_G \qquad (5)$$

## 2.3 Related Work on Private Text Embeddings

The problem of sharing privacy-preserving text embeddings under LDP has received increasing attention, confirming the relevance of our problem space. Recent notable efforts include (Du et al., 2023; Meehan et al., 2022). (Du et al., 2023) focuses on applying noise directly to sentence embeddings using DP mechanisms like Gaussian or Laplace noise, along with considering label privacy. While effective in providing LDP guarantees, its noise calibration is primarily driven by privacy parameters, in a task-agnostic manner, without explicit, learned consideration for the nuances of downstream task utility. Similarly, (Meehan et al., 2022) addresses privacy for document embeddings by generating differentially private sentence embeddings. This approach involves methods that introduce noise uniformly or based on sensitivity metrics, also operating under the LDP setting for textual representations.

These pieces of work make important contributions that operate in the same relevant setting as our work, demonstrating the feasibility and demand for LDP for textual data. Our primary novelty, which fundamentally distinguishes our method, is the task-aware calibration of noise via the NVIB. Unlike this prior work,

---

[2]These hyperparameters are unrelated to the Rényi order $\lambda$ used in the privacy definitions.

which employs task-agnostic noise mechanisms that apply noise uniformly or based on general sensitivity to individual vectors, NVDP integrates the privacy mechanism directly into the learning process of the transformer. This allows the noise level to be dynamically adjusted based on the specific utility requirements of the downstream task, optimizing the privacy-utility trade-off over the whole multi-vector document embedding. This input-dependent learned calibration ensures that sensitive information is protected adequately, while critical task-relevant features are preserved more effectively, a capability not explicitly demonstrated or achieved by the task-agnostic approaches in (Du et al., 2023; Meehan et al., 2022).

## 3 NVDP: Nonparametric Variational Differential Privacy

Our goal is to share transformer embeddings in a privacy-preserving manner, but still share embeddings which are useful. Intuitively, RDP measures the degree of privacy by measuring how much information a sampled embedding is expected to convey (with more weight put on the worst case samples). If we know what type of task we intend the embeddings to be used for, then we can measure the degree of utility of an embedding by measuring the accuracy of a model trained on such a task. Reducing the total amount of information in a latent representation while retaining enough information to perform a task is precisely the objective of a VIB regularizer. VIB trains a posterior distribution over embeddings, a sample from this posterior provides limited information, and this information is optimized to perform the downstream task. More precisely, the KL divergence used in VIB is exactly equivalent to the RD used in RDP as $\lambda$ approaches 1 (at which point all samples get equal weight, including the worst case).

We apply this general idea to the case of transformer embeddings. This is complicated by the fact that transformer embeddings consist of a sequence of vectors, so we use NVIB to provide a holistic view of the amount of information in these variable-length sequences. NVIB is used as the regularizer to calibrate the noise, and RDP is used as the privacy measure to evaluate the noise. Thus, we propose NVDP as a method to learn posterior distributions over embeddings which provide good RDP privacy and are calibrated to a given utility objective and the empirical data distribution.

### 3.1 NVDP Architecture

The architecture of our proposed NVDP model is illustrated in Figure 1. NVDP builds on top of a pre-trained transformer encoder[3], which generates multi-vector transformer embeddings. These embeddings are processed by a single layer Transformer that incorporates our modified NVIB mechanism, projecting the embeddings into a mean, variance and pseudo-count for each input vector, which together determine the posterior distribution over sequences of weighted vectors. Two key modifications are made to ensure privacy. First, we sample from this posterior distribution during training and during privatization[4] to generate a noisy, sanitized version of the embedding. This stochastic bottleneck is the core of our privacy mechanism. Second, we process these noisy embeddings with a Denoising Multi-Head Attention (MHA), but we remove the standard residual skip connection that would typically wrap this block. This architectural change is critical, as it prevents any unsanitized information from the original embedding $x$ from leaking past the noisy latent representation and into the final output. This ensures that all shared information is passed exclusively through the privacy-preserving bottleneck.

More precisely, we first map the embedding from BERT (denoted as $\boldsymbol{x} \in \mathbb{R}^{n \times d}$ in Figure 1) into the parameters $(\boldsymbol{\alpha^q}, \boldsymbol{\mu^q}, (\boldsymbol{\sigma^q})^2)$ of a Dirichlet Process $DP(\alpha_0^q, G_0^q)$, as described in Section 2.2. Rather than interpreting these parameters as specifying an exact Dirichlet Process, we interpret them as specifying a sampling procedure which stochastically generates a finite sequence of weighted vectors $\boldsymbol{S} == (\boldsymbol{\pi}, \boldsymbol{Z})$, with probability $Q(\boldsymbol{S})$. Then we run this procedure to sample one such $\boldsymbol{S}$ and share $\boldsymbol{S}$. So we can think of the posterior distribution $Q(\boldsymbol{S})$ as the embedding of the text, and think of the sample $\boldsymbol{S}$ as a noisy version of that embedding.

---

[3]We use BERT-base-uncased model to generate embeddings.

[4]We use *privatization* to refer to applying the learned stochastic mechanism to produce a noisy representation before sharing. This is distinct from downstream task inference, where the classifier makes predictions from the privatized representation.

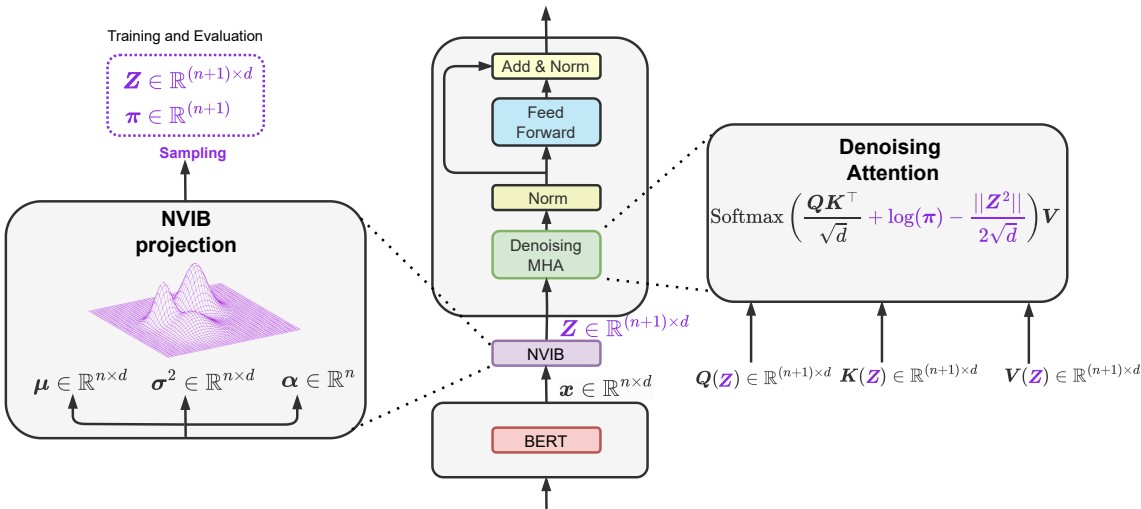

Figure 1: An NVDP model. The model projects an input embedding into the parameters of a posterior distribution. A key modification for privacy is that we sample from this latent distribution during both training and privatization. The resulting noisy representation is processed by a Denoising MHA layer. To enforce the privacy bottleneck, the standard residual skip connection around the MHA is removed, preventing any unsanitized information from bypassing the bottleneck.

## 3.2 Measuring Privacy

In our LDP setting, we want to make sure that any sample $\boldsymbol{S}$ from the embedding $Q(\boldsymbol{S})$ of a text $x$, could also have come from a different embedding $Q'(\boldsymbol{S})$ of some different text $\boldsymbol{x}'$. We measure this basic privacy criterion using the RD $D_\lambda(Q||Q')$, defined in Definition 2.1, to quantify the difference between these two sampling distributions. In this work, we measure privacy from two complementary perspectives, which differ in how we aggregate across the possible alternative inputs $\boldsymbol{x}'$. For both these measures, we report the worst case across the given input $\boldsymbol{x}$ (the average case is reported in Table 2 in Appendix A).

The first approach measures the worst case also across all alternative inputs $\boldsymbol{x}'$, as in standard RDP. We do not assume any specific notion of adjacency between examples. In our experiments, we report the maximum RD over all input pairs as the **RDP** measure (the average case is reported in Table 2 in Appendix A ).

The second approach measures an aggregation across all alternative inputs $\boldsymbol{x}'$, as is done in BDP. This measures how much an individual's output $Q$ stands out from the crowd, thereby measuring the risk of de-anonymization. As discussed in Section 2.1, we aggregate the RD values (as calculated for the first measure) across examples $\boldsymbol{x}'$ and convert to an interpretable $(\epsilon_\mu, \delta_\mu)$-style privacy loss for the **BDP** measure.

To calculate the RD between the posterior embedding $Q(\boldsymbol{S})$ for input $\boldsymbol{x}$ and an alternative embedding $Q'(\boldsymbol{S})$ for input $\boldsymbol{x}'$, we first specify a sampling procedure which approximates sampling from a Dirichlet Process, and then calculate the RD for that sampling procedure applied to $Q$ and $Q'$. This is an upper bound on the RD between the two Dirichlet Processes, since a sequence of weighted vectors is more specific than a mixture distribution, but it is appropriate for our privacy measure because it is this sequence which is actually shared. So the RD between sampling two different finite sequences of weighted vectors from two different inputs is an accurate assessment of how easily an observer can distinguish between samples from one text's embedding and samples from the other text's embedding, providing a measure of privacy.

In our framework, the noise distribution underlying $Q(\boldsymbol{S})$ is learned from the training data through the NVIB objective. The resulting privacy guarantee is therefore defined with respect to the learned stochastic mapping $Q(\boldsymbol{S})$ applied to any input, making the noise parameters dependent on the training data. This differs from classical differential privacy mechanisms with data-independent noise. However, the NVIB objective

encourages removing information irrespective of the data distribution, while the task loss encourages retaining task-relevant information, suggesting that out-of-distribution data may affect utility more than privacy.

### 3.3 Rényi Divergence between two Sampling Distributions

Since the theory of NVIB assumes that the posterior distribution which embeds the information about the input data is a Dirichlet Process, $Q = \mathrm{DP}(G_0^q, \alpha_0^q)$, we want a sampling procedure which generates finite sequences of weighted vectors which approximate sampling from this DP. For $\mathrm{DP}(G_0^q, \alpha_0^q)$, defined in equation 4, weights can be sampled using a stick breaking process parameterized by $\alpha_0^q$, and vectors can be sampled independently by first sampling a component $i$ from the mixture distribution $G_0^q$ and then sampling a vector from that component's distribution $G_i^q = \mathcal{N}(\mu_i^q, (\sigma_i^q)^2)$. But sampling from the discrete list of components $i$ makes it difficult for NVIB to do backprop through the sampling step, so instead Henderson & Fehr (2023) propose a sampling method which first samples the total weight assigned to all the vectors from a given component $i$, and then samples weighted vectors independently from the component's distribution $G_i^q$. They prove that the DP distribution in equation 4 is equivalent to such a facorised distribution as follows:

$$F \quad = \quad \sum_{i=1}^{n+1} \rho_i F_i \qquad \boldsymbol{\rho} \quad \sim \quad \mathrm{Dir}(\alpha_1^q, \ldots, \alpha_{n+1}^q) \qquad F_i \quad = \quad \mathrm{DP}(G_i^q, \alpha_i^q) \qquad (6)$$

In particular, Henderson & Fehr (2023) show that sampling the total weights $\boldsymbol{\rho_i}$ across $i$ can be done by sampling from a Dirichlet distribution parameterized by $\boldsymbol{\alpha^q}$, and the distribution of weighted vectors sampled from each component is another Dirichlet process. To generate a sample with a finite number of vectors, they assume a given bound $\kappa_i$ on the number of vectors sampled from each component $G_i^q$, and sample their weights with a symmetric Dirichlet distribution $\mathrm{Dir}(\frac{\alpha_i^q}{\kappa_i}, .^{\kappa_i}., \frac{\alpha_i^q}{\kappa_i})$. In our experiments, one vector is sampled from each component, so $\kappa_i = 1$.

The different weighted vectors sampled from a Dirichlet Process specify a mixture distribution, which is permutation invariant. However, when a sample is generated with our sampling procedure, the weighted vectors are output in a specific order. We could have a better approximation to a Dirichlet Process by randomly permuting the weighted vectors, but for simplicity we output them in the order of the tokens in the sentence. This simplifies the computation of the RD between two samples because it implies that sampled vectors for two different inputs can be aligned by their token position, which gives us an upper bound on the Dirichlet Process case, since the ordered list is more informative.[5] This gives us the following formula for calculating the RD between samples from two different input embeddings.

$$D_\lambda(\mathrm{DP}(G_0^q, \alpha_0^q) \| \mathrm{DP}(G_0^{q'}, \alpha_0^{q'})) \qquad (7)$$

$$\leq -\left( \frac{1}{\lambda-1} \log \Gamma\left(\lambda \alpha_0^q - (\lambda-1)\alpha_0^{q'}\right) + \log \Gamma(\alpha_0^{q'}) - \frac{\lambda}{\lambda-1} \log \Gamma(\alpha_0^q) \right)$$

$$+ \sum_{i=1}^{n+1} \kappa_i \left( \frac{1}{\lambda-1} \log \Gamma(\lambda \frac{\alpha_i^q}{\kappa_i} - (\lambda-1)\frac{\alpha_i^{q'}}{\kappa_i}) + \log \Gamma(\frac{\alpha_i^{q'}}{\kappa_i}) - \frac{\lambda}{\lambda-1} \log \Gamma(\frac{\alpha_i^q}{\kappa_i}) \right)$$

$$+ \sum_{i=1}^{n+1} \kappa_i \left( \frac{\lambda}{2} \left\| \frac{\boldsymbol{\mu_i^q} - \boldsymbol{\mu_i^{q'}}}{\boldsymbol{\sigma_i'}} \right\|^2 + \frac{1}{1-\lambda} \mathbf{1} \left( \log \frac{\boldsymbol{\sigma_i'}}{(\boldsymbol{\sigma_0^p})^{(1-\lambda)}(\boldsymbol{\sigma_i^q})^\lambda} \right) \right)$$

where $\boldsymbol{\sigma_i'} = \sqrt{(1-\lambda)(\boldsymbol{\sigma_i^{q'}})^2 + \lambda(\boldsymbol{\sigma_i^q})^2}$, and $\mathbf{1}$ is a vector of 1s. The full derivation is provided in Appendix B.

The RD expression in Equation 7 provides a meaningful and tractable upper bound on the divergence between two sampling distributions induced by the Dirichlet Process posteriors. This bound relies on a variational

---

[5]In our implementation, to handle the case where the two examples have different numbers of tokens, we pad the input sequences so that they all have the same length, and then assume that pad tokens have parameters $\mu_i=\mathbf{0}$, $\sigma_i=\mathbf{1}$, $\alpha_i=\mathbf{0}$. This allows us to treat all pairs of inputs as adjacent inputs for our privacy measure. Alternatively, we could have defined adjacent inputs as all having the same length, but our current approach is more appropriate for the BDP measure and still gives a meaningful privacy measure. We leave better bounds on the RD between samples from Dirichlet Processes to future work.

approximation of the posterior distribution. It also relies on additional structural assumptions to make the computation tractable. In practice, computing RD between mixture distributions is known to be challenging, and our formulation simplifies this problem by assuming an observable alignment between components. Relaxing this assumption, for example by allowing permutation-invariant matching of components or variable numbers of sampled vectors, could lead to tighter estimates of the divergence. Similarly, more advanced approximations of divergences between mixture distributions could further improve the bound. We leave these improvements to future work, as our current formulation already provides a stable and meaningful proxy for privacy, enabling us to demonstrate a strong privacy-utility trade-off in practice.

## 4 Experiments

We empirically evaluate NVDP on NLP tasks from the GLUE benchmark (Wang et al., 2018) in terms of both accuracy and privacy. This is a commonly-used benchmark for evaluating performance on a variety of natural language understanding tasks.

### 4.1 Experimental Setup

**Datasets** We evaluate the performance of NVDP on the General Language Understanding Evaluation (GLUE) (Wang et al., 2018) benchmark, which is a collection of different natural language understanding tasks including similarity and paraphrasing tasks, text classification, and natural language inference. For natural language inference, we experiment on QNLI (Rajpurkar et al., 2016) and RTE (Dagan et al., 2005). For paraphrase detection, we evaluate on MRPC (Dolan & Brockett, 2005), STS-B (Cer et al., 2017), and QQP (Iyer et al., 2017). For text classification, we evaluate on SST-2 (Socher et al., 2013).

**Base Model** We use BERTBase, which is 12 layers and 110M parameters. We use the following hyper-parameters for fine-tuning BERT: a sequence length of 512, a train batch size of 64 and evaluation batch size of 8. We use the stable variant of the Adam optimizer (Zhang et al., 2020; Mosbach et al., 2020) with the learning rate of $2e-7$ through all experiments. We use a warm-up step of 0.2.

**Baselines** We compare our method with the prior state-of-the-art baseline and previous regularization techniques:

- Baseline (Base): we use vanilla BERTBase model without any regularization technique.

- Regularization (+REG): we use Dropout (Srivastava et al., 2014)& Weight Decay (WD) (Krogh & Hertz, 1991). Dropout is a stochastic regularization technique widely used in various large-scale language models to reduce overfitting (Devlin et al., 2019; Yang, 2019; Vaswani et al., 2017). A dropout of 0.1 is applied across all layers of BERT. Another technique is WD, which helps improve generalization by penalizing large weights with a term $\frac{\lambda}{2}||\boldsymbol{y}||$ in the loss function, where $\lambda$ controls the regularization strength. For fine-tuning pretrained models, a modified WD to use $\lambda||\boldsymbol{y} - \boldsymbol{y}0||$ is used where $\boldsymbol{y}0$ represents the pretrained weights (Chelba & Acero, 2004; Daumé III, 2007). Lee et al. (2019) showed that this adjusted version of WD enhances fine-tuning for BERT, particularly on smaller datasets, compared to the traditional approach. A WD of 0.01 is applied.

**Ablations** We consider several simplified versions of our proposed model and other privacy-preserving baselines to comprehensively evaluate the advantages of NVDP. These ablations and baselines allow us to isolate the impact of NVIB's nonparametric regularization and task-aware noise calibration. One ablation (VTDP) removes the use of Bayesian nonparametrics for regularization, and two ablations provide reference models which can only share single-vector embeddings instead of multi-vector transformer embeddings.

- **Variational Transformer Differential Privacy (VTDP):** We consider a simplified version of our proposed model where the noise introduced with NVIB is replaced with noise introduced by vector-space VIB applied to each token vector independently, which we call the VTDP model. This provides a strong reference model which is the same as our NVDP model but without the holistic

regularization of the entire sequence of vectors provided by NVIB. We utilize BERTBase with an additional VIB layer to implement the information bottleneck principle outlined by Mahabadi et al. (2021). The VIB layer encodes input representations in a compressed latent space, leveraging VIB to learn a stochastic latent variable. This variable captures task-relevant information while filtering out irrelevant features, enhancing generalization. This integration enables effective fine-tuning, particularly in low-resource scenarios, by prioritizing essential information and mitigating overfitting risks. The compressed latent representation is compared to a Gaussian prior, ensuring a structured and regularized latent space. The RDP guarantee, defined between Gaussian distributions, follows the formulation in Equation 10 of Van Erven & Harremos (2014):

$$D_\lambda(\mathcal{N}(\boldsymbol{\mu_i^q}, \boldsymbol{\sigma_i^q}) || \mathcal{N}(\boldsymbol{\mu_0^{q'}}, \boldsymbol{\sigma_0^{q'}})) \tag{8}$$

$$= \frac{\lambda}{2} \left\| \frac{\boldsymbol{\mu_i^q} - \boldsymbol{\mu_0^{q'}}}{\boldsymbol{\sigma_i'}} \right\|^2 + \frac{1}{1-\lambda} \mathbf{1} \left( \log \frac{\boldsymbol{\sigma_i'}}{(\boldsymbol{\sigma_0^{q'}})^{(1-\lambda)} (\boldsymbol{\sigma_i^q})^\lambda} \right)$$

- **Variational Information Bottleneck Fixed (VIB-fixed):** VIB-fixed employs a standard Gaussian noise mechanism that injects isotropic perturbations directly into the pooled transformer embeddings. For an input $x$ with pooled embedding $\mu(x) \in \mathbb{R}^D$, the privatized representation is defined as

$$\widetilde{z}(x) = \mu(x) + \varepsilon, \qquad \varepsilon \sim \mathcal{N}(0, \sigma^2 I_D),$$

where the noise scale $\sigma$ is fixed and shared across all dimensions and inputs (we use $\sigma = 0.55$). In contrast to our NVDP model, which learns a data-adaptive and task-aware noise structure via NVIB, VIB-fixed applies a uniform and input-independent perturbation to a single vector. This serves as a transparent benchmark closely aligned with classical Gaussian DP and admits closed-form RD computations. For two adjacent inputs $x$ and $x'$, the order-$\lambda$ RD between their noisy distributions is given by Mironov (2017):

$$D_\lambda\big(\mathcal{N}(\mu(x), \sigma^2 I_D) \,\big\|\, \mathcal{N}(\mu(x'), \sigma^2 I_D)\big) = \frac{\lambda}{2\sigma^2} \|\mu(x) - \mu(x')\|_2^2. \tag{9}$$

This provides an analytically grounded reference point for comparison with NVDP, allowing us to report summary statistics of these pairwise RD across all evaluated embeddings.

- **Variational Information Bottleneck Learned (VIB-learned):** VIB-learned extends the fixed-noise mechanism by learning a data-agnostic but dimension-wise noise scale. Instead of applying a predefined isotropic perturbation, VIB-learned maintains a trainable vector of log-standard deviations $\log \boldsymbol{\sigma} \in \mathbb{R}^D$, transformed via a softplus function to ensure positivity: $\boldsymbol{\sigma} = \mathrm{softplus}(\log \boldsymbol{\sigma})$. For an input $x$ with pooled embedding $\mu(x) \in \mathbb{R}^D$, the privatized representation is:

$$\widetilde{z}(x) = \mu(x) + \boldsymbol{\sigma} \odot \varepsilon, \qquad \varepsilon \sim \mathcal{N}(0, I_D).$$

Thus, VIB-learned samples from a Gaussian distribution with mean $\mu(x)$ and diagonal covariance $\mathrm{diag}(\boldsymbol{\sigma}^2)$, where the noise magnitudes are learned globally during training but remain input-independent. This parallels the VIB-fixed mechanism while enabling dimension-wise adaptive perturbation. For two adjacent inputs $x$ and $x'$, the order-$\lambda$ RD between their privatized distributions is Mironov (2017):

$$D_\lambda\big(\mathcal{N}(\mu(x), \mathrm{diag}(\boldsymbol{\sigma}^2)) \,\big\|\, \mathcal{N}(\mu(x'), \mathrm{diag}(\boldsymbol{\sigma}^2))\big) = \frac{\lambda}{2} \left\| \frac{\mu(x) - \mu(x')}{\boldsymbol{\sigma}} \right\|_2^2 \tag{10}$$

### 4.2 Results on the GLUE Benchmark

Table 1 presents a summary of the best privacy-utility trade-off achieved by each private model, while Figure 2 visualizes the full trade-off curves using the BDP measure.

Table 1: Privacy-utility trade-off on GLUE tasks for BERT-Base. We compare our proposed **NVDP** model against non-private baselines and private ablations (**VIB-fixed**, **VIB-learned**, and **VTDP**). For each private model, we report its best-achieved utility score alongside privacy guarantees. Privacy is measured via BDP (BDP $\downarrow$) and RD (RD $\downarrow$). Lower privacy values are better. The best-performing private model per task is **bolded**.

| Dataset | Metric | Baselines (Non-Private) | | VIB-fixed (Ablation) | | | VIB-learned (Ablation) | | | VTDP (Ablation) | | | NVDP (Ours) | | |
|---|---|---|---|---|---|---|---|---|---|---|---|---|---|---|---|
| | | Base | +REG | Score | BDP$\downarrow$ | RD (max) $\downarrow$ | Score | BDP$\downarrow$ | RD (max) $\downarrow$ | Score | BDP$\downarrow$ | RD (max) $\downarrow$ | Score | BDP$\downarrow$ | RD (max) $\downarrow$ |
| MRPC | Accuracy | 81.2 | 82.4 | 82.4 | 12.58 | 2.98 | 82.2 | 11 | 2.14 | 81.1 | 11.50 | 1.20 | **83.0** | **10.70** | **0.34** |
| | F1 Score | 86.0 | 87.6 | 87.4 | 12.58 | 2.98 | 87.5 | 11 | 2.14 | 86.5 | 11.50 | 1.20 | **87.5** | **10.70** | **0.34** |
| STS-B | Pearson | 86.0 | 85.7 | 83.9 | 20.35 | 1.74 | 83.2 | **17.34** | 1.54 | 83.6 | 22.20 | 6.61 | **85.2** | 20.93 | **1.41** |
| | Spearman | 84.9 | 84.5 | 82.7 | 20.35 | 1.74 | 81.8 | **17.34** | 1.54 | 82.3 | 22.20 | 6.61 | **84.0** | 20.93 | **1.41** |
| RTE | Accuracy | 65.9 | 66.3 | **66.9** | 11.35 | 2.71 | 65.1 | **10.77** | 2.21 | 64.1 | 11.50 | 1.94 | 64.8 | 10.90 | **1.66** |
| QQP | Accuracy | 87.8 | 88.4 | 87.3 | 18.1 | 3.17 | 86.8 | 17.2 | 2.42 | 87.6 | 15.52 | 0.85 | **88.3** | **13.01** | 1.14 |
| | F1 Score | 68.4 | 69.4 | 67.4 | 18.1 | 3.17 | 67.7 | 17.2 | 2.42 | 67.4 | 15.52 | 0.85 | **68.9** | **13.01** | 1.14 |
| QNLI | Accuracy | 89.0 | 89.7 | 89 | 12.20 | 3.17 | 88.9 | **10.95** | 2.54 | 87.1 | 16.90 | 1.80 | **89.5** | 12.10 | **0.75** |
| SST-2 | Accuracy | 92.9 | 91.9 | 91.3 | 11.18 | 2.30 | 91.2 | 11.20 | 2.28 | **92.3** | **10.90** | 0.37 | 91.7 | **10.90** | **0.19** |

**Experimental Protocol** For each model, we perform five independent runs and select the best-performing run on the validation set for final evaluation on the test set. The NVDP and VIB models provide privacy by applying the same learned stochastic mapping during privatization, ensuring that only noisy sanitized embeddings are shared. The utility of these private embeddings is measured by evaluating the classifier learned during training. The results in Table 1 represent the best privacy-utility trade-off found by sweeping across a range of regularization weights $\lambda_G$ and $\lambda_D$ (as defined in Equation 5). To quantify the privacy loss shown in Table 1, we fix the Rényi order to $\lambda = 1.1$ in Equation 7. This choice is motivated by two considerations. First, the RD formulation used in Equation 7 requires $\lambda > 1$. Second, choosing $\lambda$ close to 1 makes the Rényi divergence close to the KL divergence, which is the divergence regularized during training. Thus, $\lambda = 1.1$ provides a privacy measure that remains compatible with the RD bound while staying close to the training regularizer. This measure of privacy is appropriate for use cases where expected privacy leakage is important, although it is different from traditional worst-case privacy measures. We also fix the BDP failure probability to $\delta_\mu = 10^{-5}$ in Equation 3, a commonly used small failure probability for reporting approximate DP-style guarantees, and report both the BDP and the maximum RD (RD max) across all test set pairs. Full results for all tested regularization strengths are provided in Appendix A. Importantly, our framework is not restricted to small Rényi orders, and the RD expression in Equation 7 is valid for any $\lambda > 1$. We have conducted preliminary experiments with larger $\lambda$ values and observed similar qualitative trends in the privacy-utility trade-off relative to our baselines. A more comprehensive evaluation at larger $\lambda$ would require a dedicated study and potentially adapting the training objective to better align with worst-case privacy (e.g., optimizing RD directly instead of a KL-based regularizer). We leave these extensions to future work.

**Utility and Regularization Analysis** The results in Table 1 show that the NVDP model functions as an effective regularization technique. The utility of the NVDP model is highly competitive with the non-private, regularized baseline (+REG). On some tasks, such as MRPC, NVDP achieves a higher accuracy (83.0%) compared to both +REG (82.4%) and the private ablations VIB-fixed (82.4%) and VIB-learned (82.2%). In contrast, the VTDP ablation generally struggles to maintain the same level of utility (e.g., 81.1% on MRPC). While VIB-fixed and VIB-learned occasionally offer strong utility scores—outperforming VTDP on tasks like MRPC and RTE—they typically fail to match the consistent performance of NVDP. For instance, on STS-B, NVDP scores 85.2, surpassing VIB-fixed (83.9) and VIB-learned (83.2). Furthermore, where the baselines do achieve high utility, they often incur a higher privacy cost (higher RD values) compared to NVDP.

**Privacy-Utility Trade-off Analysis** We analyze this ability of our models to balance utility against privacy, to get a more complete understanding of the results from Table 1. Figure 2 plots accuracy against

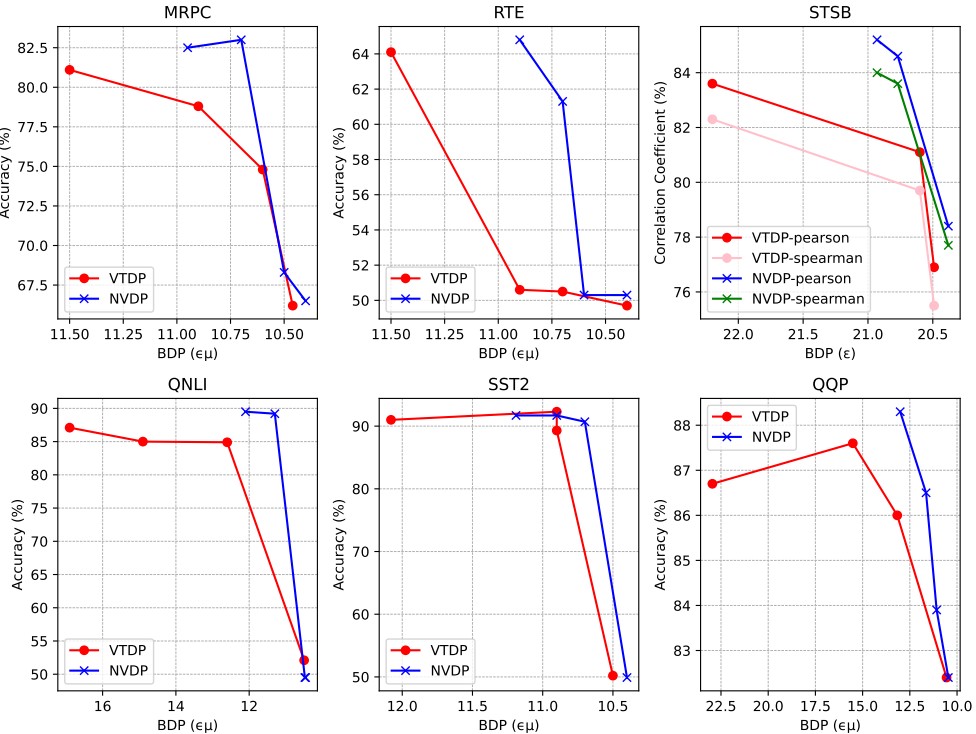

Figure 2: Accuracy versus BDP ($\epsilon_\mu$). The BDP budget ($\epsilon_\mu$) is calculated by finding the tightest privacy guarantee for a fixed $\delta_\mu = 1e - 5$. This provides a more interpretable view of the privacy-utility trade-off, where lower ($\epsilon_\mu$) values signify stronger, more practical privacy guarantees. The NVDP model consistently achieves better privacy-utility points than the VTDP ablation.

the BDP measure for the tunable models (NVDP and VTDP); detailed results are given in Table 2 in Appendix A. For this BDP guarantee, the results show a clear advantage for our proposed model. Across all datasets, the NVDP models (blue curves) consistently occupy the most favorable region of the plot – closest to the top-right corner – indicating a superior trade-off compared to the VTDP ablation (red curves). For example, on MRPC, to achieve a privacy budget comparable to NVDP (e.g., $\approx 10.6$), the VTDP ablation's accuracy drops to just 74.8%.

This superior performance is further highlighted when comparing against the private ablations VIB-fixed and VIB-learned reported in Table 1. While these baselines offer competitive utility, they incur a significantly higher privacy cost, outside the scale shown in Figure 2. For instance, on MRPC, NVDP reaches 83.0% accuracy with a BDP($\epsilon_\mu$) of 10.70. In contrast, VIB-fixed achieves a lower accuracy (82.4%) yet requires a much looser privacy budget (12.58). This underscores the superior privacy-utility frontier of our NVDP method, which is a direct consequence of the task-aware randomness calibration via NVIB. This distinguishes it from the task-agnostic noise injection strategies employed by the baselines and other existing LDP mechanisms (Du et al., 2023; Meehan et al., 2022), suggesting that NVIB is more effective at removing extraneous information while retaining utility.

This conclusion is strongly reinforced by the RD values in Table 1. As a direct measure of distinguishability, the RD reveals a substantial gap between NVDP and the baselines. In the same MRPC comparison, NVDP's worst-case RD is 0.34. Conversely, VIB-fixed and VIB-learned exhibit much higher information leakage, with RD values of 2.98 and 2.14, respectively – nearly an order of magnitude higher. The benefit is particularly clear on SST-2; while NVDP achieves an RD of just 0.19, both VIB-fixed and VIB-learned exceed 2.20,

and the VTDP ablation sits at 0.37. These RD results confirm that NVDP provides the tightest privacy guarantees for a given level of utility.

Together, these results demonstrate that NVDP is uniquely capable of preserving task-critical information while maintaining tight privacy guarantees. Its use of learned dynamic noise levels is much better than task-agnostic noise injection strategies at achieving good privacy guarantees while maintaining high utility. Its use of NVIB to manage the holistic noise level over an entire sequence shows consistent improvement over managing noise levels independently for each vector in a transformer's multi-vector embeddings.

## 5 Conclusion

We propose a model that addresses the privacy concerns in sharing data for deep learning by integrating a NVIB layer into the transformer architecture and sharing its transformer embeddings. By sharing embeddings sampled from the NVIB layer, our proposed NVDP model controls the amount of shared information despite there being many vectors in each transformer embedding, and this information is tailored to the needs of the target downstream task. We quantify the level of privacy with RD DP, and show that our NVDP model is able to share embeddings of data which provide strong privacy guarantees while maintaining good downstream model utility.

Our experimental results demonstrate the effectiveness of this approach by evaluating the privacy-utility trade-off. We confirmed that NVDP consistently controls information leakage more effectively than a strong VIB-based ablation across a range of GLUE tasks. Crucially, by converting our RD measurements into interpretable $(\epsilon_\mu, \lambda_\mu)$-BDP guarantees, we have shown that our model can achieve strong, practical privacy budgets while maintaining high model utility. This is a significant step towards deploying privacy-preserving transformer embeddings in real-world applications where clear and meaningful privacy assurances are required.

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

## A    Additional Privacy Analysis

This appendix provides supplementary results to complement the analysis in the main text. Table 2 presents the comprehensive numerical results presented in Table 1, detailing the utility and privacy metrics for all tested KL divergence regularization weights in equation 5. So, this table provides the complete data used to generate the summary results in Table 1. Figure 2 visualizes the privacy-utility trade-offs for the worst case *max* RD. These results demonstrate that our NVDP model consistently outperforms the VTDP ablation under all evaluation settings.

**Effect of regularization parameters.** In our experiments, we explicitly vary the KL regularization weights $\lambda_G$ and $\lambda_D$ from Equation 5, using the values $\{10^{-3}, 10^{-2}, 10^{-1}, 1\}$ for both parameters (with $\lambda_G = \lambda_D$ for NVDP and VTDP). These parameters directly control the strength of the information bottleneck by weighting the KL divergence terms in the loss.

Increasing $\lambda_G$ and $\lambda_D$ strengthens the information bottleneck constraint, reducing the mutual information between inputs and latent representations. As a result, the learned posterior distributions become less distinguishable across inputs, leading to lower RD values and tighter BDP guarantees (lower $\epsilon_\mu$).

Empirically, Table 2 shows that as $\lambda_G$ and $\lambda_D$ increase, both RD and BDP values consistently decrease, reflecting improved privacy. However, this comes at the cost of reduced task utility, illustrating the expected privacy-utility trade-off.

**Analysis of QNLI and QQP (Figure 3 versus Figure 2).** While the BDP-based results in the main text show a consistent advantage for NVDP, Figure 3 reveals some negative trends when considering the worst-case RD. In particular, for datasets such as QQP and QNLI, stronger regularization settings (e.g., $(\lambda_G, \lambda_D) \in \{0.1, 1\}$) can lead to a decrease in utility compared to VTDP. This behavior arises because RD captures the maximum divergence across all example pairs, making it highly sensitive to rare or difficult examples. In contrast, BDP reflects an aggregation over the data distribution and therefore captures the expected privacy risk. As a result, NVDP maintains a better overall privacy-utility trade-off under BDP, even in cases where worst-case RD highlights isolated failures.

This suggests that a small number of challenging examples may require stronger or adaptive noise control. Incorporating mechanisms such as example-wise information clipping or adaptive noise scaling could further improve worst-case guarantees, which we leave for future work.

**Improved performance on STS-B (Table 2).** For the STS-B dataset, we observe that increasing the regularization strength (e.g., from $\lambda_G = \lambda_D = 10^{-1}$ to 1) for NVDP can improve both privacy and utility. We attribute this behavior to the denoising and smoothing effect of the information bottleneck. As a regression task, STS-B benefits from continuous and structured representations. Moderate regularization suppresses irrelevant variations in the embedding space, improving generalization while simultaneously reducing information leakage.

This highlights that NVDP can act not only as a privacy mechanism but also as a task-aware regularizer. In particular, tasks that rely on fine-grained semantic similarity can benefit from this smoothing effect, resulting in simultaneous gains in utility and privacy.

Table 2: Full experimental results for BERT-Base on GLUE, showing the complete privacy-utility trade-off across all tested KL divergence regularization weights ($\lambda_G, \lambda_D \in \{10^{-3}, 10^{-2}, 10^{-1}, 1\}$). For each model, we report Utility (task-specific scores) and privacy metrics BDP ($\epsilon_\mu$) and RD. The RD metric are reported as max/avg, representing the worst-case and average-case divergence across all example pairs, respectively. Lower values are better for all privacy metrics ($\downarrow$).

| Model | Metric | MRPC | STS-B | RTE | QQP | QNLI | SST-2 |
|---|---|---|---|---|---|---|---|
| | | Acc / F1 | Pearson / Spearman | Accuracy | Acc / F1 | Accuracy | Accuracy |
| BERT$_{\text{Base}}$ | Utility | 81.2 / 86.0 | 86.0 / 84.9 | 65.9 | 87.8 / 68.4 | 89.0 | 92.9 |
| | Privacy | - | - | - | - | - | - |
| +REG | Utility | 82.4 / 87.6 | 85.7 / 84.5 | 66.3 | 88.4 / 69.4 | 89.7 | 91.9 |
| | Privacy | - | - | - | - | - | - |
| VIB-fixed | Utility | 82.4 / 87.4 | 83.9 / 82.7 | 66.9 | 87.3 / 67.4 | 89.0 | 91.3 |
| | BDP ($\epsilon_\mu$) $\downarrow$ | 12.58 | 20.35 | 11.35 | 18.1 | 12.20 | 11.18 |
| | RD (max/avg) $\downarrow$ | 2.98 / 0.30 | 1.74 / 0.14 | 2.71 / 0.14 | 3.17 / 0.42 | 3.17 / 0.30 | 2.30 / 0.38 |
| VIB-learned | Utility | 82.2 / 87.5 | 83.2 / 81.8 | 65.1 | 86.8 / 67.7 | 88.9 | 91.2 |
| | BDP ($\epsilon_\mu$) $\downarrow$ | 11.00 | 17.34 | 10.77 | 17.2 | 10.95 | 11.20 |
| | RD (max/avg) $\downarrow$ | 2.14 / 0.30 | 1.54 / 0.14 | 2.21 / 0.31 | 2.42 /0.41 | 2.54 / 0.23 | 2.28 / 0.23 |
| NVDP$_{\lambda_G=\lambda_D=1e-3}$ | Utility | 82.5 / 87.1 | 85.2 / 84.0 | 64.8 | 88.3 / 68.9 | 89.5 | 91.7 |
| | BDP ($\epsilon_\mu$) $\downarrow$ | 10.95 | 20.93 | 10.90 | 13.01 | 12.10 | 11.19 |
| | RD (max/avg) $\downarrow$ | 0.89 / 0.06 | 1.41 / 0.10 | 1.66 / 0.12 | 1.143 / 0.031 | 0.75 / 0.06 | 1.00 / 0.06 |
| NVDP$_{\lambda_G=\lambda_D=1e-2}$ | Utility | 83.0 / 87.5 | 84.6 / 83.6 | 61.3 | 86.5 / 67.9 | 89.2 | 91.7 |
| | BDP ($\epsilon_\mu$) $\downarrow$ | 10.70 | 20.77 | 10.70 | 11.64 | 11.30 | 10.90 |
| | RD (max/avg) $\downarrow$ | 0.34 / 0.02 | 1.22 / 0.05 | 0.87 / 0.04 | 0.53 / 0.028 | 0.71 / 0.09 | 0.19 / 0.01 |
| NVDP$_{\lambda_G=\lambda_D=1e-1}$ | Utility | 68.3 / 80.2 | 78.4 / 77.7 | 50.3 | 83.9 / 65.5 | 49.5 | 90.7 |
| | BDP ($\epsilon_\mu$) $\downarrow$ | 10.50 | 20.38 | 10.60 | 11.08 | 10.48 | 10.70 |
| | RD (max/avg) $\downarrow$ | 0.04 / 0.01 | 0.22 / 0.01 | 0.10 / 0.01 | 0.37 / 0.02 | 0.016 / 0.003 | 0.016 / 0.004 |
| NVDP$_{\lambda_G=\lambda_D=1}$ | Utility | 66.5 / 79.9 | 82.7 / 82.9 | 50.3 | 82.4 / 0 | 49.5 | 49.9 |
| | BDP ($\epsilon_\mu$) $\downarrow$ | 10.40 | 18.65 | 10.40 | 10.46 | 10.46 | 10.40 |
| | RD (max/avg) $\downarrow$ | 0.008 / 0.002 | 0.03 / 0.004 | 0.005 / 0 | 0.006 / 0.001 | 0.007 / 0.001 | 0.01 / 0.002 |
| VTDP$_{\lambda_G=\lambda_D=1e-3}$ | Utility | 81.1 / 86.5 | 83.6 / 82.3 | 64.1 | 86.7 / 67.6 | 87.1 | 91.0 |
| | BDP ($\epsilon_\mu$) $\downarrow$ | 11.50 | 22.20 | 11.50 | 22.95 | 16.90 | 12.08 |
| | RD (max/avg) $\downarrow$ | 1.2 / 0.37 | 6.61 / 0.77 | 1.94 / 0.44 | 3.33 / 0.59 | 1.80 / 0.68 | 1.45 / 0.54 |
| VTDP$_{\lambda_G=\lambda_D=1e-2}$ | Utility | 78.8 / 84.9 | 81.1 / 79.7 | 50.6 | 87.6 / 67.4 | 85.0 | 92.3 |
| | BDP ($\epsilon_\mu$) $\downarrow$ | 10.90 | 20.60 | 10.90 | 15.52 | 14.90 | 10.90 |
| | RD (max/avg) $\downarrow$ | 0.43 / 0.12 | 1.33 / 0.13 | 1.62 / 0.33 | 0.85 / 0.14 | 0.39 / 0.14 | 0.37 / 0.15 |
| VTDP$_{\lambda_G=\lambda_D=1e-1}$ | Utility | 74.8 / 83.3 | 76.9 / 75.5 | 50.5 | 86.0 / 62.3 | 84.9 | 89.3 |
| | BDP ($\epsilon_\mu$) $\downarrow$ | 10.60 | 20.49 | 10.70 | 13.16 | 12.60 | 10.90 |
| | RD (max/avg) $\downarrow$ | 0.049 / 0.018 | 0.33 / 0.04 | 0.18 / 0.04 | 0.09 / 0.019 | 0.04 / 0.019 | 0.13 / 0.05 |
| VTDP$_{\lambda_G=\lambda_D=1}$ | Utility | 66.2 / 79.6 | 52.6 / 51.6 | 49.7 | 82.4 / 0.3 | 52.1 | 50.2 |
| | BDP ($\epsilon_\mu$) $\downarrow$ | 10.46 | 20.30 | 10.40 | 10.55 | 10.50 | 10.50 |
| | RD (max/avg) $\downarrow$ | 0 / 0 | 0.038 / 0.005 | 0 / 0 | 0 / 0 | 0 / 0 | 0 / 0 |

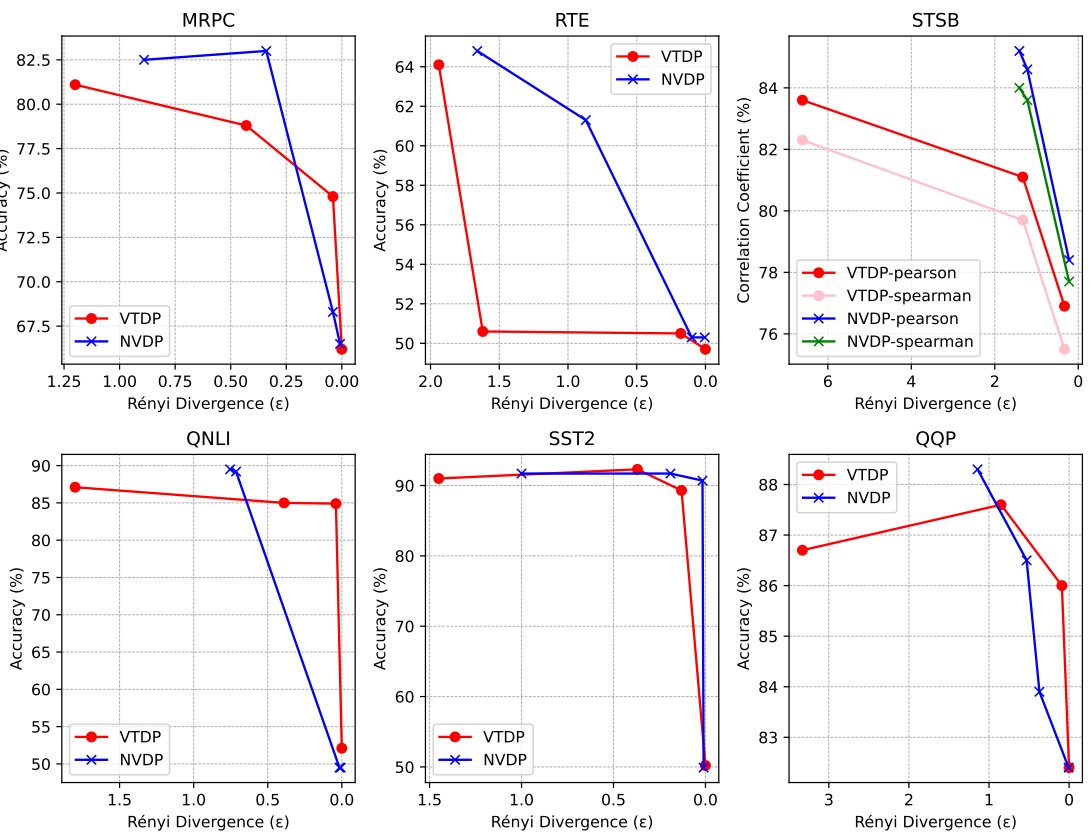

Figure 3: Accuracy versus **maximum** RD illustrating the worst-case privacy-utility trade-off. Each point corresponds to a different KL regularization weight, where stronger regularization leads to better privacy (lower RD). The most favorable models are those closest to the upper-right corner.

# B  Deriving the Rényi Divergence between Dirichlet Distributions

Here we provide the proof that:

$$D_\lambda(\mathrm{DP}(G_0^q, \alpha_0^q) \| \mathrm{DP}(G_0^{q'}, \alpha_0^{q'})) \tag{11}$$

$$\leq - \left( \frac{1}{\lambda-1} \log \Gamma\left(\lambda \alpha_0^q - (\lambda-1)\alpha_0^{q'}\right) + \log \Gamma(\alpha_0^{q'}) - \frac{\lambda}{\lambda-1} \log \Gamma(\alpha_0^q) \right)$$

$$+ \sum_{i=1}^{n+1} \kappa_i \left( \frac{1}{\lambda-1} \log \Gamma(\lambda \frac{\alpha_i^q}{\kappa_i} - (\lambda-1)\frac{\alpha_i^{q'}}{\kappa_i}) + \log \Gamma(\frac{\alpha_i^{q'}}{\kappa_i}) - \frac{\lambda}{\lambda-1} \log \Gamma(\frac{\alpha_i^q}{\kappa_i}) \right)$$

$$+ \sum_{i=1}^{n+1} \kappa_i \left( \frac{\lambda}{2} \left\| \frac{\boldsymbol{\mu}_i^q - \boldsymbol{\mu}_i^{q'}}{\boldsymbol{\sigma}_i'} \right\|^2 + \frac{1}{1-\lambda} \mathbf{1} \left( \log \frac{\boldsymbol{\sigma}_i'}{(\boldsymbol{\sigma}_i^{q'})^{(1-\lambda)}(\boldsymbol{\sigma}_i^q)^\lambda} \right) \right)$$

where $\boldsymbol{\sigma}_i' = \sqrt{(1-\lambda)(\boldsymbol{\sigma}_i^{q'})^2 + \lambda(\boldsymbol{\sigma}_i^q)^2}$, $\mathbf{1}$ is a vector of 1s, and $\kappa_i$ is the number of vectors generated from component $G_i^q$.

Let $\boldsymbol{F} \sim \mathrm{DP}(G_0^q, \alpha_0^q)$ and $\boldsymbol{F'} \sim \mathrm{DP}(G_0^{q'}, \alpha_0^{q'})$ be finite samples from the Dirichlet processes corresponding to two input embeddings, defined as:

$$
\begin{array}{rclcrclcrcl}
\alpha_0^q & = & \displaystyle\sum_{i=1}^{n+1} \alpha_i^q \; ; & & G_0^q & = & \displaystyle\sum_{i=1}^{n+1} \frac{\alpha_i^q}{\alpha_0^q} G_i^q & ; G_i^q & = & \mathcal{N}(\boldsymbol{\mu}_i^q, \boldsymbol{I}(\boldsymbol{\sigma}_i^q)^2) & (12) \\[4mm]
\alpha_0^{q'} & = & \displaystyle\sum_{i=1}^{n+1} \alpha_i^{q'} \; ; & & G_0^{q'} & = & \displaystyle\sum_{i=1}^{n+1} \frac{\alpha_i^{q'}}{\alpha_0^{q'}} G_i^{q'} & ; G_i^{q'} & = & \mathcal{N}(\boldsymbol{\mu}_i^{q'}, \boldsymbol{I}(\boldsymbol{\sigma}_i^{q'})^2). & (13)
\end{array}
$$

Sampling both the total weights for each component $G_i^q$, and sampling the weights across vectors from a given component, are Dirichlet distributions, thus:

$$
\begin{aligned}
\boldsymbol{F} & = \sum_{i=1}^{n+1} \rho_i F_i & (14) \\
\boldsymbol{\rho} & \sim \mathrm{Dir}(\alpha_1^q, \ldots, \alpha_{n+1}^q) \\
F_i & = \mathrm{DP}(G_i^q, \alpha_i^q)
\end{aligned}
$$

Let $dp(\boldsymbol{F};\ (\boldsymbol{G}^q, \boldsymbol{\alpha}^q))$ (where $\boldsymbol{G}^q = (G_1^q, \ldots, G_{n+1}^q)$ and $\boldsymbol{\alpha}^q = (\alpha_1^q, \ldots, \alpha_{n+1}^q)$) be the probability density function for $\boldsymbol{F} \sim \mathrm{DP}(G_0^q, \alpha_0^q)$. Similarly, let $fdp(\boldsymbol{\rho}, \boldsymbol{F};\ (\boldsymbol{G}^q, \boldsymbol{\alpha}^q))$ be the probability density function for $\boldsymbol{\rho} \sim \mathrm{Dir}(\alpha_1^q, \ldots, \alpha_{n+1}^q)$ and $F_i \sim \mathrm{DP}(G_i^q, \alpha_i^q)$, and let $d(\boldsymbol{\rho};\ \boldsymbol{\alpha}^q)$ be the probability density function for $\boldsymbol{\rho} \sim \mathrm{Dir}(\alpha_1^q, \ldots, \alpha_{n+1}^q)$.

Given this setup, the RD between the Dirichlet processes factorises into a $\boldsymbol{\rho}$ term and a $\boldsymbol{F}$ term:

$$
D_\lambda(\mathrm{DP}(G_0^q, \alpha_0^q) \| \mathrm{DP}(G_0^{q'}, \alpha_0^{q'})) \tag{15}
$$

$$
= \frac{1}{\lambda - 1} \log\left( \int_{(\boldsymbol{F})} \frac{(dp(\boldsymbol{F};\ (\boldsymbol{G}^q, \boldsymbol{\alpha}^q)))^\lambda}{(dp(\boldsymbol{F};\ (\boldsymbol{G}^{q'}, \boldsymbol{\alpha}^{q'})))^{\lambda-1}} \right)
$$

$$
\leq \frac{1}{\lambda - 1} \log\left( \int_{(\boldsymbol{\rho}, \boldsymbol{F})} \frac{(fdp(\boldsymbol{\rho}, \boldsymbol{F};\ (\boldsymbol{G}^q, \boldsymbol{\alpha}^q)))^\lambda}{(fdp(\boldsymbol{\rho}, \boldsymbol{F};\ (\boldsymbol{G}^{q'}, \boldsymbol{\alpha}^{q'})))^{\lambda-1}} \right)
$$

$$
= \frac{1}{\lambda - 1} \log\left( \int_{(\boldsymbol{\rho}, \boldsymbol{F})} \frac{\left(d(\boldsymbol{\rho};\ \alpha_1^q, \ldots, \alpha_{n+1}^q) \prod_i dp(F_i;\ G_i^q, \alpha_i^q)\right)^\lambda}{\left(d(\boldsymbol{\rho};\ \alpha_1^{q'}, \ldots, \alpha_{n+1}^{q'}) \prod_i dp(F_i;\ G_i^{q'}, \alpha_i^{q'})\right)^{\lambda-1}} \right)
$$

$$
= \frac{1}{\lambda - 1} \log\left( \int_{\boldsymbol{\rho}} \frac{\left(d(\boldsymbol{\rho};\ \alpha_1^q, \ldots, \alpha_{n+1}^q)\right)^\lambda}{\left(d(\boldsymbol{\rho};\ \alpha_1^{q'}, \ldots, \alpha_{n+1}^{q'})\right)^{\lambda-1}} \right)
$$

$$
+ \sum_{i=1}^{n+1} \frac{1}{\lambda - 1} \log\left( \int_{F_i} \frac{(dp(F_i;\ G_i^q, \alpha_i^q))^\lambda}{\left(dp(F_i;\ G_i^{q'}, \alpha_i^{q'})\right)^{\lambda-1}} \right)
$$

$$
= D_\lambda(\mathrm{Dir}(\alpha_1^q, \ldots, \alpha_{n+1}^q) \| \mathrm{Dir}(\alpha_1^{q'}, \ldots, \alpha_{n+1}^{q'})) \tag{16}
$$

$$
+ \sum_{i=1}^{n+1} D_\lambda(\mathrm{DP}(G_i^q, \alpha_i^q) \| \mathrm{DP}(G_i^{q'}, \alpha_i^{q'})). \tag{17}
$$

In a similar way, we now want to factorize Equation 17, $D_\lambda(\mathrm{DP}(G_i^q, \alpha_i^q) || \mathrm{DP}(G_i^{q'}, \alpha_i^{q'}))$:

For each $i$, let $\kappa_i$ be the number of $z_k$ generated from $G_i^q$ and $\boldsymbol{\pi}$ be the weights of these vectors, then:

$$D_\lambda(\mathrm{DP}(G_i^q, \alpha_i^q) || \mathrm{DP}(G_i^{q'}, \alpha_i^{q'})) \tag{18}$$

$$= \frac{1}{\lambda - 1} \log\left(\int_{(\boldsymbol{\pi}, \boldsymbol{Z})} \frac{\left(d(\boldsymbol{\pi}; \frac{\alpha_i^q}{\kappa_i}, \overset{\kappa_i}{\ldots}, \frac{\alpha_i^q}{\kappa_i}) \prod_k g(\boldsymbol{z}_k; \boldsymbol{\mu}_i^q, \boldsymbol{\sigma}_i^q)\right)^\lambda}{\left(d(\boldsymbol{\pi}; \frac{\alpha_i^{q'}}{\kappa_i}, \overset{\kappa_i}{\ldots}, \frac{\alpha_i^{q'}}{\kappa_i}) \prod_k g(\boldsymbol{z}_k; \boldsymbol{\mu}_i^{q'}, \boldsymbol{\sigma}_i^{q'})\right)^{\lambda-1}}\right)$$

$$= \frac{1}{\lambda - 1} \log\left(\int_{\boldsymbol{\pi}} \frac{\left(d(\boldsymbol{\pi}; \frac{\alpha_i^q}{\kappa_i}, \overset{\kappa_i}{\ldots}, \frac{\alpha_i^q}{\kappa_i})\right)^\lambda}{\left(d(\boldsymbol{\pi}; \frac{\alpha_i^{q'}}{\kappa_i}, \overset{\kappa_i}{\ldots}, \frac{\alpha_i^{q'}}{\kappa_i})\right)^{\lambda-1}}\right)$$

$$+ \sum_{k=1}^{\kappa_i} \frac{1}{\lambda - 1} \log\left(\int_{\boldsymbol{z}_k} \frac{(g(\boldsymbol{z}_k; \boldsymbol{\mu}_i^q, \boldsymbol{\sigma}_i^q))^\lambda}{\left(g(\boldsymbol{z}_k; \boldsymbol{\mu}_i^{q'}, \boldsymbol{\sigma}_i^{q'})\right)^{\lambda-1}}\right)$$

$$= D_\lambda(\mathrm{Dir}(\frac{\alpha_i^q}{\kappa_i}, \overset{\kappa_i}{\ldots}, \frac{\alpha_i^q}{\kappa_i}) || \mathrm{Dir}(\frac{\alpha_i^{q'}}{\kappa_i}, \overset{\kappa_i}{\ldots}, \frac{\alpha_i^{q'}}{\kappa_i}))$$

$$+ \kappa_i D_\lambda(\mathcal{N}(\boldsymbol{\mu}_i^q, \boldsymbol{\sigma}_i^q) || \mathcal{N}(\boldsymbol{\mu}_i^{q'}, \boldsymbol{\sigma}_i^{q'}))$$

We can thus combine all the above RD of Equation 15 into a single RD for the original DPs, for a given $\boldsymbol{\kappa}$ specifying all $\kappa_i$. This computation is exact for the given $\kappa_i$.

$$D_\lambda(\mathrm{DP}(G_0^q, \alpha_0^q) || \mathrm{DP}(G_0^{q'}, \alpha_0^{q'})) \tag{19}$$

$$\leq D_\lambda(\mathrm{Dir}(\alpha_1^q, \ldots, \alpha_{n+1}^q) || \mathrm{Dir}(\alpha_1^{q'}, \ldots, \alpha_{n+1}^{q'})) \tag{20}$$

$$+ \sum_{i=1}^{n+1} D_\lambda(\mathrm{Dir}(\frac{\alpha_i^q}{\kappa_i}, \overset{\kappa_i}{\ldots}, \frac{\alpha_i^q}{\kappa_i}) || \mathrm{Dir}(\frac{\alpha_i^{q'}}{\kappa_i}, \overset{\kappa_i}{\ldots}, \frac{\alpha_i^{q'}}{\kappa_i})) \tag{21}$$

$$+ \sum_{i=1}^{n+1} \kappa_i D_\lambda(\mathcal{N}(\boldsymbol{\mu}_i^q, \boldsymbol{\sigma}_i^q) || \mathcal{N}(\boldsymbol{\mu}_i^{q'}, \boldsymbol{\sigma}_i^{q'})). \tag{22}$$

Now we can derive a bound for each of these terms of the RD.

**For the first Dirichlet case, Equation 20**, using the formula in Table 2 from Gil et al. (2013), we have:

$$D_\lambda(\text{Dir}(\alpha_1^q, \ldots, \alpha_{n+1}^q) \| \text{Dir}(\alpha_1^{q'}, \ldots, \alpha_{n+1}^{q'})) \tag{23}$$

$$= \frac{1}{\lambda - 1} \log(\frac{\beta(\alpha^{q'})^{(\lambda-1)}}{\beta(\alpha^q)^{(\lambda)}} \beta(\lambda \alpha^q - (\lambda - 1)\alpha^{q'}))$$

$$\text{if for all } i, \ \alpha_i^q + (\lambda - 1)(\alpha_i^q - \alpha_i^{q'}) > 0 \tag{24}$$

$$= \frac{1}{\lambda - 1} \log \left( \frac{\prod_{i=1}^{n+1} \Gamma(\alpha_i^{q'})}{\Gamma(\sum_{i=1}^{n+1} \alpha_i^{q'})} \right)^{(\lambda-1)} - \frac{1}{\lambda - 1} \log \left( \frac{\prod_{i=1}^{n+1} \Gamma(\alpha_i^q)}{\Gamma(\sum_{i=1}^{n+1} \alpha_i^q)} \right)^\lambda$$

$$+ \frac{1}{\lambda - 1} \log \left( \frac{\prod_{i=1}^{n+1} \Gamma\left(\lambda \alpha_i^q - (\lambda - 1)\alpha_i^{q'}\right)}{\Gamma\left(\lambda \sum_{i=1}^{n+1} \alpha_i^q - (\lambda - 1) \sum_{i=1}^{n+1} \alpha_i^{q'}\right)} \right)$$

$$= \sum_{i=1}^{n+1} \log \Gamma(\alpha_i^{q'}) - \log \Gamma(\alpha_0^{q'}) - \frac{\lambda}{\lambda - 1} \left( \sum_{i=1}^{n+1} \log \Gamma(\alpha_i^q) - \log \Gamma(\alpha_0^q) \right)$$

$$+ \frac{1}{\lambda - 1} \sum_{i=1}^{n+1} \log \Gamma\left(\lambda \alpha_i^q - (\lambda - 1)\alpha_i^{q'}\right) - \frac{1}{\lambda - 1} \log \Gamma\left(\lambda \alpha_0^q - (\lambda - 1)\alpha_0^{q'}\right)$$

$$= - \left( \frac{1}{\lambda - 1} \log \Gamma\left(\lambda \alpha_0^q - (\lambda - 1)\alpha_0^{q'}\right) + \log \Gamma(\alpha_0^{q'}) - \frac{\lambda}{\lambda - 1} \log \Gamma(\alpha_0^q) \right)$$

$$+ \sum_{i=1}^{n+1} \left( \frac{1}{\lambda - 1} \log \Gamma\left(\lambda \alpha_i^q - (\lambda - 1)\alpha_i^{q'}\right) + \log \Gamma(\alpha_i^{q'}) - \frac{\lambda}{\lambda - 1} \log \Gamma(\alpha_i^q) \right)$$

**For the second Dirichlet case 21**, we have:

$$D_\lambda(\text{Dir}(\frac{\alpha_i^q}{\kappa_i}, \overset{\kappa_i}{\cdots}, \frac{\alpha_i^q}{\kappa_i})|| \text{Dir}(\frac{\alpha_i^{q'}}{\kappa_i}, \overset{\kappa_i}{\cdots}, \frac{\alpha_i^{q'}}{\kappa_i})) \tag{25}$$

$$= \frac{1}{\lambda-1} \log(\frac{\beta(\frac{\alpha_i^{q'}}{\kappa_i})^{(\lambda-1)}}{\beta(\frac{\alpha_i^q}{\kappa_i})^{(\lambda)}} \beta(\lambda\frac{\alpha_i^q}{\kappa_i} - (\lambda-1)\frac{\alpha_i^{q'}}{\kappa_i}))$$

$$\text{if for all } i, \frac{\alpha_i^q}{\kappa_i} + (\lambda-1)(\frac{\alpha_i^q}{\kappa_i} - \frac{\alpha_i^{q'}}{\kappa_i}) > 0 \tag{26}$$

$$= \frac{1}{\lambda-1} \log \left( \frac{\prod_1^{\kappa_i} \Gamma(\frac{\alpha_i^{q'}}{\kappa_i})}{\Gamma(\sum_1^{\kappa_i} \frac{\alpha_i^{q'}}{\kappa_i})} \right)^{(\lambda-1)} - \frac{1}{\lambda-1} \log \left( \frac{\prod_1^{\kappa} \Gamma(\frac{\alpha_i^q}{\kappa_i})}{\Gamma(\sum_1^{\kappa_i} \frac{\alpha_i^q}{\kappa_i})} \right)^{\lambda}$$

$$+ \frac{1}{\lambda-1} \log \left( \frac{\prod_1^{\kappa_i} \Gamma\left(\lambda\frac{\alpha_i^q}{\kappa_i} - (\lambda-1)\frac{\alpha_i^{q'}}{\kappa_i}\right)}{\Gamma\left(\lambda\sum_1^{\kappa_i} \frac{\alpha_i^q}{\kappa_i} - (\lambda-1)\sum_1^{\kappa_i} \frac{\alpha_i^{q'}}{\kappa_i}\right)} \right)$$

$$= \log \Gamma(\frac{\alpha_i^{q'}}{\kappa_i})^{\kappa_i} - \log \Gamma(\alpha_i^{q'}) - \frac{\lambda}{\lambda-1} \left( \log \Gamma(\frac{\alpha_i^q}{\kappa_i})^{\kappa_i} - \log \Gamma(\alpha_i^q) \right)$$

$$+ \frac{1}{\lambda-1} \left( \log \Gamma(\lambda\frac{\alpha_i^q}{\kappa_i} - (\lambda-1)\frac{\alpha_i^{q'}}{\kappa_i})^{\kappa_i} - \log \Gamma(\lambda\alpha_i^q - (\lambda-1)\alpha_i^{q'}) \right)$$

$$= - \left( \frac{1}{\lambda-1} \log \Gamma(\lambda\alpha_i^q - (\lambda-1)\alpha_i^{q'}) + \log \Gamma(\alpha_i^{q'}) - \frac{\lambda}{\lambda-1} \log \Gamma(\alpha_i^q) \right)$$

$$+ \kappa_i \left( \frac{1}{\lambda-1} \log \Gamma(\lambda\frac{\alpha_i^q}{\kappa_i} - (\lambda-1)\frac{\alpha_i^{q'}}{\kappa_i}) + \log \Gamma(\frac{\alpha_i^{q'}}{\kappa_i}) - \frac{\lambda}{\lambda-1} \log \Gamma(\frac{\alpha_i^q}{\kappa_i}) \right)$$

**For the Gaussian term 22:**

For the Gaussian term, for all $i$, let:

$$\boldsymbol{\sigma}_i' = \sqrt{(1-\lambda)(\boldsymbol{\sigma}_i^{q'})^2 + \lambda(\boldsymbol{\sigma}_i^q)^2} > 0 \tag{27}$$

and then in reference to equation 10 in (Van Erven & Harremos, 2014):

$$D_\lambda(\mathcal{N}(\boldsymbol{\mu}_i^q, \boldsymbol{\sigma}_i^q)||\mathcal{N}(\boldsymbol{\mu}_i^{q'}, \boldsymbol{\sigma}_i^{q'})) = \frac{\lambda}{2}\left\|\frac{\boldsymbol{\mu}_i^q - \boldsymbol{\mu}_i^{q'}}{\boldsymbol{\sigma}_i'}\right\|^2 + \frac{1}{1-\lambda}\mathbf{1}\left(\log \frac{\boldsymbol{\sigma}_i'}{(\boldsymbol{\sigma}_i^{q'})^{(1-\lambda)}(\boldsymbol{\sigma}_i^q)^\lambda}\right) \tag{28}$$

**Finally, substituting into equation 19, we get the final value for RD:**

$$D_\lambda(\mathrm{DP}(G_0^q, \alpha_0^q) \| \mathrm{DP}(G_0^{q'}, \alpha_0^{q'})) \tag{29}$$

$$\leq -\left( \frac{1}{\lambda-1} \log\Gamma\left(\lambda\alpha_0^q - (\lambda-1)\alpha_0^{q'}\right) + \log\Gamma(\alpha_0^{q'}) - \frac{\lambda}{\lambda-1}\log\Gamma(\alpha_0^q) \right)$$

$$+ \sum_{i=1}^{n+1}\left( \frac{1}{\lambda-1}\log\Gamma\left(\lambda\alpha_i^q - (\lambda-1)\alpha_i^{q'}\right) + \log\Gamma(\alpha_i^{q'}) - \frac{\lambda}{\lambda-1}\log\Gamma(\alpha_i^q) \right)$$

$$- \sum_{i=1}^{n+1}\left( \frac{1}{\lambda-1}\log\Gamma(\lambda\alpha_i^q - (\lambda-1)\alpha_i^{q'}) + \log\Gamma(\alpha_i^{q'}) - \frac{\lambda}{\lambda-1}\log\Gamma(\alpha_i^q) \right)$$

$$+ \sum_{i=1}^{n+1}\kappa_i\left( \frac{1}{\lambda-1}\log\Gamma(\lambda\frac{\alpha_i^q}{\kappa_i} - (\lambda-1)\frac{\alpha_i^{q'}}{\kappa_i}) + \log\Gamma(\frac{\alpha_i^{q'}}{\kappa_i}) - \frac{\lambda}{\lambda-1}\log\Gamma(\frac{\alpha_i^q}{\kappa_i}) \right)$$

$$+ \sum_{i=1}^{n+1}\kappa_i\left( \frac{\lambda}{2}\left\|\frac{\boldsymbol{\mu}_i^q - \boldsymbol{\mu}_i^{q'}}{\boldsymbol{\sigma}_i'}\right\|^2 + \frac{1}{1-\lambda}\mathbf{1}\left(\log\frac{\boldsymbol{\sigma}_i'}{(\boldsymbol{\sigma}_i^{q'})^{(1-\lambda)}(\boldsymbol{\sigma}_i^q)^\lambda}\right) \right)$$

$$= -\left( \frac{1}{\lambda-1}\log\Gamma\left(\lambda\alpha_0^q - (\lambda-1)\alpha_0^{q'}\right) + \log\Gamma(\alpha_0^{q'}) - \frac{\lambda}{\lambda-1}\log\Gamma(\alpha_0^q) \right)$$

$$+ \sum_{i=1}^{n+1}\kappa_i\left( \frac{1}{\lambda-1}\log\Gamma(\lambda\frac{\alpha_i^q}{\kappa_i} - (\lambda-1)\frac{\alpha_i^{q'}}{\kappa_i}) + \log\Gamma(\frac{\alpha_i^{q'}}{\kappa_i}) - \frac{\lambda}{\lambda-1}\log\Gamma(\frac{\alpha_i^q}{\kappa_i}) \right)$$

$$+ \sum_{i=1}^{n+1}\kappa_i\left( \frac{\lambda}{2}\left\|\frac{\boldsymbol{\mu}_i^q - \boldsymbol{\mu}_i^{q'}}{\boldsymbol{\sigma}_i'}\right\|^2 + \frac{1}{1-\lambda}\mathbf{1}\left(\log\frac{\boldsymbol{\sigma}_i'}{(\boldsymbol{\sigma}_i^{q'})^{(1-\lambda)}(\boldsymbol{\sigma}_i^q)^\lambda}\right) \right)$$

The respective constraints for each RD are:

From constraint 24 of the first Dirichlet case, we get:

$$\alpha_i^q + (\lambda-1)(\alpha_i^q - \alpha_i^{q'}) > 0$$

$$\lambda\alpha_i^q > (\lambda-1)\alpha_i^{q'}$$

$$\frac{\lambda}{\lambda-1} > \frac{\alpha_0^{q'}}{\alpha_0^q}$$

Constraint 26 leads to:

$$\frac{\alpha_i^q}{\kappa_i} + (\lambda-1)(\frac{\alpha_i^q}{\kappa_i} - \frac{\alpha_i^{q'}}{\kappa_i}) > 0$$

$$\lambda\frac{\alpha_i^q}{\kappa_i} > (\lambda-1)\frac{\alpha_i^{q'}}{\kappa_i}$$

$$\frac{\lambda}{\lambda-1} > \frac{\alpha_0^{q'}}{\alpha_0^q}$$

Constraint 27 leads to:

$$\sqrt{(1-\lambda)(\boldsymbol{\sigma}_i^{q'})^2 + \lambda(\boldsymbol{\sigma}_i^q)^2} > 0$$

$$(1-\lambda)(\boldsymbol{\sigma}_i^{q'})^2 + \lambda(\boldsymbol{\sigma}_i^q)^2 > 0$$

$$\lambda(\boldsymbol{\sigma}_i^q)^2 > (\lambda-1)(\boldsymbol{\sigma}_i^{q'})^2$$

$$\frac{\lambda}{\lambda-1} > \frac{(\boldsymbol{\sigma}_i^{q'})^2}{(\boldsymbol{\sigma}_i^q)^2}$$

Therefore, all share the same constraint $\frac{\lambda}{\lambda-1} > \frac{\alpha_0^{q'}}{\alpha_0^q}$ and one constraint that applies only if $\kappa_i > 0$: $\frac{\lambda}{\lambda-1} > \frac{(\boldsymbol{\sigma}_i^{q'})^2}{(\boldsymbol{\sigma}_i^q)^2}$.

