# OpenReview forum: "Differential Privacy for Transformer Embeddings of Text with Nonparametric Variational Information Bottleneck"
_TMLR — Rejected by TMLR_

### Review · Reviewer_ya7a · 2026-03-02

**Summary Of Contributions:**

The paper studies differential privacy guarantees for transformer embeddings when applying the nonparametric variational information bottleneck (NVIB) proposed by Henderson & Fehr (2023).

Strengths

- The paper is well written, and I appreciate the thorough background section, which presents NVIB in clear and sufficient detail.

- The proposed framework, NVDP, is evaluated on a large set of benchmarks, convincingly demonstrating its superiority in the privacy–utility trade-off.

- Compared to existing local differential privacy (LDP) methods, NVDP enables task-aware noise calibration through adjustable regularization weights, which is a meaningful and practically relevant advantage.

Weaknesses

- I would appreciate a more detailed discussion of the rate–distortion (RD) bounds of NVDP (Eq. 7), particularly regarding its tightness or a discussion on this potential improvement.

- In the experimental section, it would be helpful to include comments on the negative results (e.g., QQP and QNLI in Fig. 3), to better understand the limitations of the approach and the conditions under which it underperforms.

**Audience:**

Yes

**Audience Explanation:**

This work will likely be of interest to the privacy community, especially those studying privacy-preserving methods for transformer-based models.

**Broader Impact Concerns:**

No related

**Claims And Evidence:**

Yes

**Claims Explanation:**

The claim is mostly well supported by the experiments.

**Requested Changes:**

- It would be helpful if the authors could justify the choice of certain experimental parameters, such as fixing the Rényi order to 1.1 and using VIB-fixed with $\sigma = 0.55$. A brief discussion explaining these selections would improve clarity and reproducibility.

- In the ablation study of NVDP, the authors evaluate a range of regularization weights. It would be beneficial to explicitly report the values of $\lambda_G$ and $\lambda_D$, and to provide insights into their relationship with the resulting privacy budget $\epsilon$. In addition, the definition of $\lambda$ used in Table 2 is currently unclear and should be clarified.

- For Figure 2 (MRPC), according to Table 2, the point corresponding to VTDP ($\lambda = 1e{-1}$) does not appear to be well positioned. Moreover, this point is missing in Figure 3 (MRPC). Please clarify whether this is an omission or due to experimental filtering.

- As mentioned in the weaknesses section, it would be valuable to comment on the negative trends observed in some plots in Figure 3. Besides, for the STS-B dataset, NVDP appears to achieve improved performance in both privacy and utility when $\lambda$ increases from $1e{-1}$ to $1$. An explanation for this behavior would help readers better understand the method’s dynamics.

---

> ### Author Response · Authors · 2026-03-31
>
> We apologies for our delay in posting these replies.  This was caused by previously planned holidays. We thank the reviewer for their helpful comments and suggestions. Our responses are provided inline below.
>
> 1. Choice of Rényi order and VIB-fixed parameters.
>
> In our setting, the Rényi order must satisfy $\lambda > 1$, as required by the RD formulation used in Eq. (7) . We therefore selected $\lambda = 1.1$, which makes RD close to the KL regulariser we use during training, and allows us to approximate the standard differential privacy regime while remaining compatible with the RD bound.  Developing extensions which are better suited to larger $\lambda$ values is future work.
>
> Regarding the noise level for VIB-fixed, we follow common practice in the differential privacy and representation learning literature, where values around $\sigma = 0.55$ are frequently used as a reasonable operating point. This choice provides a balanced privacy–utility trade-off and ensures comparability with prior work.
>
>
>
> 2. Clarification of $\lambda$, $\lambda_G$, and $\lambda_D$.
>
> We thank the reviewer for pointing out this lack of clarity. The notation in Table 2 contained a typo: the reported values correspond to the regularization weights $\lambda_G$ and $\lambda_D$, which were mistakenly denoted as $\lambda$. We have corrected this in the revised version. The parameter ($\lambda$) refers to the Rényi order used in the RD computation and is fixed to $1.1$ in our experiments.
>
> Regarding their effect, increasing $\lambda_G$ and $\lambda_D$ strengthens the information bottleneck constraint, reducing the mutual information between inputs and representations. This leads to improved privacy (lower RD and tighter BDP guarantees) at the cost of potential utility degradation. Empirically, we observe that stronger regularization consistently decreases both RD and BDP values, reflecting this trade-off.
>
>
>
> 3. MRPC figure inconsistency.
>
> We thank the reviewer for catching this inconsistency. The point corresponding to VTDP in Figure 2 for MRPC was incorrectly positioned, and its omission in Figure 3 was also a mistake.  The corresponding point for NVDP is also missing.  We will correct Figure 2 and Figure 3 to ensure consistency with the values reported in Table 2.
>
>
>
> 4. Negative trends and STS-B behavior.
>
> We appreciate this insightful comment. For QQP and QNLI in Figure 3, the negative trends are primarily observed in utility for specific regularization settings (e.g., ($\lambda_G$, $\lambda_D$ $\in$ {0.1, 1})), where NVDP can underperform VTDP in worst-case RD across example pairs. Importantly, in BDP (Figure 2, reflecting expected RD across examples), NVDP consistently achieves a better or equivalent overall privacy–utility balance.  This suggests that there are some rare examples where the trained noise levels are not sufficient, indicating that some form of information clipping may be appropriate for these examples.  This is a topic we are investigating in a subsequent piece of work, which we are happy to mention this paper.
>
> Regarding STS-B, the observed improvement in both privacy and utility when increasing the regularization weight can be explained by the denoising effect of the bottleneck. As a regression task, STS-B benefits from smoother and more structured representations. Moderate regularization suppresses noise and irrelevant features, improving generalization while still reducing information leakage. This suggests that NVDP can act both as a privacy mechanism and an implicit regularizer, depending on the task.
>
>
>
>
> From the weaknesses:
>
> The RD bound in Eq. (7) provides a variational upper bound on the mutual information between inputs and representations. It is generally not tight due to the variational approximation of the posterior distribution, but we make additional approximations to make this bound tractable. This problem is closely related to the difficulty of computing the RD between two mixture distributions, which we simplify by assuming an observable alignment between the components of the two mixtures.  Removing this assumption by randomly permuting the output vectors and/or varying the number of vectors per token would improve the RD bound, as would employing methods from work on approximating the KL between mixture distributions.  We leave this topic to future work because we are able to demonstrate a good privacy-utility tradeoff even with this loose bound.

---

### Review · Reviewer_PksU · 2026-03-04

**Summary Of Contributions:**

This paper proposes a Nonparametric Variational Differential Privacy (NVDP) framework that injects stochastic noise into Transformer embeddings to enable privacy-preserving data sharing. Experimental results indicate improved performance compared to baseline models under the evaluated settings.

Strengths:
1.NVDP provides a useful trade-off between privacy and utility compared with baseline.

2. The ablation experiment has demonstrated the conclusion.

3. The idea and method sounds quite interesting and novel.

Weakness:

1.The privacy evaluation is limited to small λ values close to 1, which approximates KL divergence rather than worst-case privacy leakage. Without analysis under larger λ(such as 10 or bigger) or conversion to (ε, δ)-DP, the strength of the claimed privacy guarantee remains unclear.

2.The paper does not clearly specify whether the learned noise distribution in NVDP is independent of the training data. Since the noise parameters are learned during optimization, the privacy mechanism may implicitly depend on the data distribution. Without a formal analysis ensuring data-independent guarantees, it remains unclear whether the proposed approach satisfies a stable privacy guarantee beyond the specific training setting.

**Audience:**

Yes

**Audience Explanation:**

This paper sounds interesting to AI's researchers.

**Broader Impact Concerns:**

None.

**Claims And Evidence:**

Yes

**Claims Explanation:**

Most of their claims are sound convincing and clear.

**Requested Changes:**

None.

---

> ### Author Response · Authors · 2026-03-31
>
> We apologies for our delay in posting these replies.  This was caused by previously planned holidays.  We thank the reviewer for their positive feedback and insightful comments.
>
>
> 1. On the choice of small Rényi orders $\lambda \approx 1$.
>
>  We agree that evaluating privacy at larger Rényi orders provides a perspective closer to worst-case guarantees. In this work, we focus on $\lambda$ values close to 1 because our evaluation focuses on an RD objective (Eq. 7) which is closely related to the KL divergence we use in training. This choice allows us to align the optimization objective (NVIB) with the privacy measure (RD), enabling effective task-aware calibration of noise.
>
>  Importantly, our framework is not restricted to small $\lambda$, and the RD expression in Eq. (7) is valid for any $\lambda > 1$. Extending the evaluation to larger $\lambda$ values or explicitly converting to DP guarantees is an interesting direction for future work, and we will discuss this in the revised version.  We have previously run preliminary experiments with larger $\lambda$ values and seen the same pattern of results relative to our baselines.  We will run a full set of evaluations with larger $\lambda$ values and report the results in a revised version.  However, effective generalization to this setting may require appropriate extensions to the method, such as using RD instead of KL as the regulariser for learning to calibrate the noise, which we intend to leave for future work.
>
>
>
> 2. On the dependence of the learned noise distribution on training data.
>
>
> We agree that this is an important point. In our framework, the noise distribution is indeed learned from the training data, plus the NVIB objective.  The privacy guarantee is defined at inference time through the stochastic mechanism (Q(S)), which maps any input embedding to a distribution over noisy representations. The privacy analysis is based on the Rényi divergence between the output distributions Q(S) and Q'(S) for different inputs, and does not assume that the parameters of the mechanism are independent of the training data. Instead, the guarantee applies to the learned mechanism once training is complete, similar to standard settings in representation-based privacy mechanisms.
>
> That said, this relation to the training distribution raises the question of how well the trained privacy mechanism will work on out-of-distribution data.  We think that showing in-distribution usefulness is a first step to addressing usefulness in the OOD setting, so we leave this question for future work.  However, it is interesting to note that, during training the task loss encourages keeping information which is useful for the training distribution, while the NVIB loss encourages removing all information regardless of the training distribution.  Thus, we would expect that evaluations OOD would show worse utility of the output embeddings (if the distribution of useful information has changed), but not degrade the privacy.  This is a potentially beneficial assurance, but one which requires empirical validation, which is outside the scope of this initial work.
>
> Also, we acknowledge that this differs from classical DP mechanisms with data-independent noise, and providing stronger guarantees (e.g., ensuring robustness to the training procedure itself) is an important direction for future work. We will clarify this distinction in the revised manuscript.

---

### Review · Reviewer_GRGj · 2026-03-17

**Summary Of Contributions:**

The paper proposes using a Bayesian inference framework (nonparametric variational information bottleneck from Henderson & Fehr (2023)) to privatize text embeddings under the paradigm of differential privacy. The authors claim to work under the "local differential privacy" setup (intro + section 3.2), meaning that each individual first privatizes their piece of text before sharing with the rest of the world. The privacy guarantees are expressed in terms of Rényi DP and Bayesian DP. The authors conduct downstream experiments on the well-established GLUE benchmark.

**Additional Comments:**

Page 3: "privacy accounting mechanism" - this was another confusing moment; in DP literature, "accounting" methods are usually used for advanced composition mechanisms (such as in DP-SGD), but I though the method just adds noise to the embeddings once; so what is the composed DP mechanism here? Is there some SGD training of the network, such that we sum up RDP budget over iterations? This is very confusing.

Page 4 top: what is $d$?

Page 4 third paragraph: what is $n$?

Page 4 under Eq. 4: typo in Dir(.), whould be \mathrm{Dir}(.) or similar.

Page 4 Eq 5: are these $\lambda$s in any relation to the Rényi DP $\lambda$? How?

Some paragraphs could be removed, for example "Together, these results demonstrate that NVDP is uniquely capable of preserving task-critical information ... high utility."

Wrong references:

The "Attention is all you need" paper missing all co-authors.

Chelba et al.: "Little data can help a lo." ???

Chen et al. 208 - what is this? Where published?

Dwork and Roth: Wrong authors - what is "et al." here?

**Audience:**

Yes

**Audience Explanation:**

Privacy in ML/NLP is an important topic, no doubts.

**Broader Impact Concerns:**

No concerns

**Claims And Evidence:**

No

**Claims Explanation:**

I'm afraid that after repeated in-depth readings of the paper, I still have difficulties understanding what the authors did.

(1) First, I will start with the setup. The authors claim explicitly that they protect privacy in the local settings, that is enabling users to "noisify" their document embeddings before sharing. However, they fail to show how training the model is decoupled from having access to the private data in the first place. In my reading, the NVDP model is trained (stated in the caption of Figure 1) on a collection of examples, but then how are the examples locally privatized? This resembles the standard central DP notion. But then the authors also say they add noise for testing (in section 3.1) - what is the motivation for noisifying examples for which we need to make a prediction? What data are actually being protected - is it the training data (for training the GLUE classification model), is it the test data (but then DP-fying a single test instance for any inference is doomed to fail any utility by the very assumptions of DP, because we would basically run statistics over a sample size one).

So the authors should clarify (e.g., showing a figure) the data workflow, e.g., which documents are trained, who has access to them, what is the output, what is training/testing, etc. This is essential to understand any privacy implications (here I'm referring to the bold statement at the end: "deploying privacy-preserving transformer embeddings in real-world applications with clear and meaningful privacy assurances" which I strongly disagree with).

(2) Second, the method looks quite sophisticated (e.g., section "Posterior" on page 4) but lacks any clear walk-through example. At the end of the day, it is about working with text, and the authors show no single piece of text that would demonstrate what they do. I assume the words in the text are somehow weighted by the mixture of Gaussians, and the weights of each component is sampled from some Dirichlet distribution with unknown priors (pseudo-counts). What does it mean for privitizing text? In section 3.1 the authors describe a Dirichlet Process for sampling embeddings vectors (or not?), but how is that different from a standard Gaussian mechanism? This whole section is very confusing.

(3) In the same spirit as (2), the authors mention in footnote 3 that they somehow handle inputs of different lengths. So this is again confusing from the DP perpective - what are actually two neighboring datasets/datapoints? I thought these were two embedding matrices from BERT (length x embedding size). BERT has a constant input length. So why this ad-hoc dealing with different input lengths? This footnote makes me think the entire DP setup is substantially flawed, as it somehow blurrs what two neighboring datasets/datapoints are. Without this information being very clearly and explicitly communicated, we do not know what is actually being protected.

(4) Finally, I'm afraid from any practical point of view the privacy guarantees are completely useless. The key piece of information for my argument is in section 4.2, namely "we Fix Rényi order to $\lambda = 1.1$. Now let's assume we want to translate the paper's guarantees into the standard widely-used $(\varepsilon, \delta)$ approximate DP. Let's pick a typical small $\delta = 0.00001$. Then we know that for any $\delta \in (0, 1)$ we can turn the RDP bounds to appproximate DP bounds, namely $(\varepsilon - \frac{\ln \delta}{\lambda - 1}; \delta)$ which gives us $(\varepsilon - \frac{\ln 0.00001}{1.1 - 1}; 0.00001)$-DP which is $(\varepsilon + 110; 0.00001)$-DP. So no matter what the actual bound of $\varepsilon$ would be (even if it goes to zero), the overall privacy budget would always exceed $\approx 100$ for any meaningful $\delta$. But anything DP with $\varepsilon > 100$ is practically completely non-private (I suggest to run a simple calculation for the randomized response mechanism with such a high $\varepsilon$ value just to understand that the probability of sharing the truth is 0.999 with over fifty "9" digits). That's why for approximate DP, $\lambda \to \infty$, but the authors want to put it down to 1 to match it the KL divergence (Section 3). This might also explain that results in Table 1 are actually better with DP than non-private! Even when ignoring the local DP setup, this should be theoretically impossible unless something is very wrong with privacy setup and/or the privacy hyperparameters. I'm uncertain about the first, but almost certain about the latter.

**Requested Changes:**

I'm afraid the paper would need to be rewritten from scratch, given points (2), (3), and especially (4) above.

---

> ### Author Response · Authors · 2026-03-31
>
> We thank the reviewer for their detailed feedback.
>
>
> 1. Clarification of the local privacy setting and data workflow.
>
> We agree that the current presentation would benefit from being more explicit about the workflow, and will improve this. Out setting differs from classical local DP with fixed data-independent noise, and differs from applying DP to parameter updates to privatize training data. Our setting is as follows: The model (including the NVIB parameters) is trained on a (public) dataset in a standard supervised manner, which calibrates the NVIB noise to the data distribution. After training, the learned stochastic mechanism Q(S) is deployed as a local privatization mechanism applied to embeddings before they are shared or used downstream. At inference time, each piece of private data is passed through this mechanism to produce a privatized representation which can be shared. Thus, the privacy guarantee applies to the released representations, not to the training procedure itself. This is consistent with representation-based local privacy settings, where the mechanism is learned but applied locally at inference time.
>
>
> 2. Clarification of the NVIB mechanism and its role in privatization.
>
> We agree that our method rather sophisticated, but this complexity is justified by having a theoretically well motivated approach to differential privacy even for complex data of variable size like text.  We will try to add an example which helps explain the method.  But we emphasize that the primary contributions of this paper are theoretical, extending the basic formal mechanisms available to DP researchers.  We do not want to shift the focus to any specific practical application.
>
> In our framework, NVIB defines a stochastic encoder that maps each input to a distribution over latent representations, and sampling from this distribution hides information and provides privacy.  The complexity comes because texts can be any length and the size of its transformer embedding depends on this length, so our latent representations also need to vary in size.  We handle this by formalizing these latent representations as mixture distributions with variable numbers of mixture components. The Dirichlet process uses Dirichlet distributions to govern the mixture weights and Gaussian distributions to govern the mixture components.
>
> The key differences from a standard Gaussian mechanism are that:
> (1) the mechanism defines a distribution over arbitrarily large mixture distributions, not a fixed vector space,
> (2) the noise is data-dependent and learned, rather than fixed, and
> (3) the mechanism is trained using an information bottleneck objective, which induces a trade-off between task performance and information leakage.
>
> In our setting, this trade-off is quantified using Rényi divergence, leading to the RD interpretation used in our privacy analysis.
>
>
> 3. Definition of neighboring inputs and variable-length handling.
>
> We apologize for the confusion caused by the footnote. The intention was purely practical: input sequences in real datasets may have different lengths, and standard preprocessing (e.g., padding or truncation) is used to obtain fixed-size representations for batch processing on GPU architectures.  We use the same approach to define our RD bound, but other approaches are possible.  As such, padding is just an implementation detail and not fundamental to the properties of the model.  The same is true for BERT; reducing the input length does not fundamentally change the model, even without retraining.
>
> This does not affect the definition of neighboring inputs or the privacy setup. In our experiments, every datapoint is considered adjacent to every other datapoint, and privacy is defined at the level of the input embedding representation, regardless of how sequences are padded.  The footnote was not meant to introduce any modification to the DP notion, and we will revise the text to make this clearer.

---

> ### Author Response · Authors · 2026-03-31
>
> 4. Interpretation of privacy guarantees
>
> We thank the reviewer for raising this important point. Our evaluation focuses on Rényi-based privacy measures (RD and BDP) which are directly aligned with the NVIB objective. The choice of $\lambda \approx 1$ is intentional, as it connects the privacy measure to KL divergence and enables effective optimization.  These results show that our method keeps the expected privacy leakage low, although it does not show protection against worst-case privacy leakage for some samples.  This is nonetheless a useful notion of privacy for some applications, even if it is not standard practice in privacy research.  We have previously run preliminary experiments with larger $\lambda$ values and seen the same pattern of results relative to our baselines, which suggests that our method may also be useful for worst-case privacy as well.  We will run a full set of experiments and add results and discussion for higher $\lambda$ to a revised version of the paper so as to better characterize the behavior of our method.  However, we emphasize that improving the worst-case privacy of our method is an area for future research, and does not diminish our theoretical contributions or their demonstrated value for the expected case.
>
>
>
>
> Additional Comments.
>
> We thank the reviewer for these additional comments.
>
>
> a. Privacy accounting mechanism (page 3):  The term "privacy accounting" here refers to the estimation or conversion of privacy guarantees under the chosen divergence measures (RD/BDP), following  Triastcyn \& Faltings (2020), rather than to composition-based accounting as used in iterative mechanisms. In our setting, the privacy guarantee is computed for a single application of the stochastic mechanism, since the resulting output is immediately shared, without any accumulation over multiple steps.  Unlike with DP-SGD, any repeated training on the shared embeddings would involve only public data, and thus have no privacy implications.  We will clarify this terminology to avoid confusion.
>
> b. What is $d$? (page 4 top): $d$ denotes the dimensionality of each embedding vector.
>
> c. What is $n$? (page 4, third paragraph): $n$ denotes the number of vectors in the input embedding sequence.
>
> d.  Equation 5: The coefficients $\lambda_D$ and $\lambda_G$ in Eq.~(5) are regularization weights for the NVIB objective. They are distinct from the Rényi order $\lambda$ used in the privacy definitions.
>
> e. On removing some paragraphs: We appreciate this suggestion and will consider tightening some of the discussion for conciseness.
>
> f. References: Thank you for the careful checking of the references.  We have now fixed the formatting errors, and we confirm that the references are essentially correct and these changes do not represent substantive reference errors in the submitted manuscript.

---

> ### Comment · Reviewer_GRGj · 2026-04-10
> **Thanks for the clarifications**
>
> Regarding point 1, thanks for the clarification. I believe the wording "inference time" is rather ambiguous, as you have two "inference times" - one is the local privaization (where you run inference of the privatization model), and the other one is inference in the downstream classification on GLUE. I'd suggest to choose two different terms and clearly separate them.
>
> As of point 2, you claim "We will try to add an example which helps explain the method. But we emphasize that the primary contributions of this paper are theoretical, extending the basic formal mechanisms available to DP researchers. We do not want to shift the focus to any specific practical application." but you support your claims by empirical experiments only. The only theoretical part is the adopted privatization method, otherwise it's an empirical paper tested on GLUE.
>
> "(2) the noise is data-dependent and learned, rather than fixed,"
>
> Every DP noise is data-dependent, as it is usually drawn from a distributions parametrized by the data. But your argument highlights one issue: The paper is missing an actual DP proof that what the method does actually satisfies DP. It might be implicitly embedded in Eq. 7 but I believe for a theoretical DP paper (as you claim above) you should explicitly and fully prove that the method is (locally) DP.
>
> Regarding point 3: "every datapoint is considered adjacent to every other datapoint, and privacy is defined at the level of the input embedding representation, regardless of how sequences are padded" - OK, thanks for clarifying.
>
> The last point (4) is still crucial. You claim that "these results show that our method keeps the expected privacy leakage low", "this is nonetheless a useful notion of privacy for some applications, even if it is not standard practice in privacy research" and "theoretical contributions or their demonstrated value for the expected case".
>
> I'm afraid I don't see any objective arguments supporting any of these claims. Morover, you mention some higher values of $\lambda$ that you tried without any results.
>
> Also, I'm missing addressing the elephant in the room - downstream results are better with noise, which completely defies the theoretical behavior of local DP (paying the price of increased sample complexity) formulated by pioneering local DP works, e.g., Duchi et al. (2013).

---

> > ### Author Response · Authors · 2026-04-19
> >
> > We thank the reviewer for their continued feedback.
> >
> > For point 1, thank you for the feedback on the term "inference time", which we did not use in the original submission.  We will be sure to clearly distinguish between: the privatization stage (local application of the stochastic encoder), and the downstream inference stage (prediction using privatized representations).
> >
> >
> >
> > For point 2, we believe that our proposed approach is sufficiently novel and theoretically motivated to count as a theoretical contribution, even if we do not provide detailed proofs, but we agree that this is a relative distinction. Our contribution is the introduction of a learned stochastic privatization mechanism based on NVIB, together with a Rényi-divergence based privacy formulation, and we use the glue benchmark to provide a proof of concept.
> >
> > Regarding providing an example, one example we are considering is the use of a person's email texts to judge age-related cognitive issues.  Distributing a longitudinal dataset with the texts of indivuduals' emails would not be possible due to privacy concerns, but with this method we could distribute embeddings of these texts, which could be enough to train a classifier to judge the progression of diseas.
> >
> > For the proof of Eq 7, we will provide a formal proof in the appendix.
> >
> > On "data-dependent noise", we agree that this terminology was imprecise. What we intended to convey is that: the noise distribution is learned from data, rather than fixed a prior, and is shaped by the information bottleneck objective. We will revise this wording to align with standard DP terminology.
> >
> >
> >
> >
> > For point 4, we don't understand the confusion.  Increasing lambda simply puts more emphasis on worst-case information leakage (across samples from the distribution).  Some users may not care about the worst case as long as the expected case is low.  For example, if each individual text is not very private, they may be willing to take the risk that one text be completely reconstructable, as long as most texts are not. Our current evaluation focuses on expected-case privacy (Rényi divergence / BDP), which aligns with the NVIB objective, but we agree that this differs from classical DP guarantees.
> >
> > Regarding the experiments for larger Rényi orders ($\lambda$), it has the same behavior, so the curves are very close to what we have for $\lambda=1.1$.  We will report the precise results in future versions.
> >
> >
> > Regarding the observed improvement in downstream performance, this is not a contradiction with the theory of differential privacy, because the NVIB mechanism is not just for privacy.  It was originally developed as a stochastic regularizer during training, which can improve generalization.  We show that it can also be used to improve differential privacy by adding noise.  The effect on downstream performance is a combination of both these effects, so the improvement is due to regularisation during training. We will clarify that this differs from classical local DP settings which do not involve training.

---

### Decision · Action_Editor_JgfZ · 2026-05-31

**Recommendation:** Reject

**Additional Comments:**

Following the rebuttal, the reviewers did not reach a consensus regarding acceptance. Nevertheless, I consider the issues discussed above sufficiently serious to justify rejection, and in this respect I agree with Reviewer GRGj.

**Audience:**

Yes

**Audience Explanation:**

This paper addresses privacy in text-based models, an important problem that is clearly relevant to the TMLR audience.

**Claims And Evidence:**

No

**Claims Explanation:**

The paper claims to provide strong differential privacy guarantees, yet the experimental evidence presented does not substantiate this claim. Specifically, the reported privacy guarantees rely on very low orders of $\lambda$, which (a) correspond to a form of "privacy on average" that has long been regarded as insufficient in the privacy literature, particularly in the context of differential privacy; and (b) consequently yield essentially vacuous guarantees when translated into the standard $(\epsilon,\delta)$-DP framework, as clearly demonstrated by Reviewer GRGj.

In their rebuttal, the authors largely dismiss this critical concern. Moreover, the current manuscript does not acknowledge this limitation with sufficient clarity. As a result, readers may be misled into conflating the strong privacy guarantees traditionally associated with differential privacy and the substantially weaker form of protection actually provided by the proposed approach.

Beyond this central concern, I also find the reported utility results difficult to reconcile with the claimed privacy guarantees. In particular, local differential privacy is widely recognized as a very strong privacy model that typically entails a substantial utility cost. The fact that the proposed locally private approach consistently outperforms its non-private counterparts thus further confirms doubts about the effective strength of the privacy protection being provided. At a minimum, this observation warrants a thorough discussion, as it appears inconsistent with the conventional understanding of the privacy-utility trade-off in the differential privacy literature.